# A sequential EMT-MET mechanism drives the differentiation of human embryonic stem cells towards hepatocytes

Qiuhong Li[1,2,3,*], Andrew P. Hutchins[1,*], Yong Chen[1], Shengbiao Li[1], Yongli Shan[1], Baojian Liao[1], Dejin Zheng[1], Xi Shi[1], Yinxiong Li[1], Wai-Yee Chan[1,4], Guangjin Pan[1], Shicheng Wei[2,3], Xiaodong Shu[1] & Duanqing Pei[1]

Reprogramming has been shown to involve EMT–MET; however, its role in cell differentiation is unclear. We report here that in vitro differentiation of hESCs to hepatic lineage undergoes a sequential EMT–MET with an obligatory intermediate mesenchymal phase. Gene expression analysis reveals that Activin A-induced formation of definitive endoderm (DE) accompanies a synchronous EMT mediated by autocrine TGFβ signalling followed by a MET process. Pharmacological inhibition of TGFβ signalling blocks the EMT as well as DE formation. We then identify SNAI1 as the key EMT transcriptional factor required for the specification of DE. Genetic ablation of *SNAI1* in hESCs does not affect the maintenance of pluripotency or neural differentiation, but completely disrupts the formation of DE. These results reveal a critical mesenchymal phase during the acquisition of DE, highlighting a role for sequential EMT–METs in both differentiation and reprogramming.

[1] CAS Key laboratory of Regenerative Biology, Guangdong Key laboratory of Stem Cell and Regenerative Medicine and CUHK-GIBH Joint Research Laboratory on Stem Cells and Regenerative Medicine, Guangzhou Institute of Biomedicine and Health-Guangzhou Medical University Joint School of Biological Sciences, South China Institute of Stem Cell Biology and Regenerative Medicine, Guangzhou Institutes of Biomedicine and Health, Chinese Academy of Sciences, 190 Kai Yuan Avenue, Science Park, Guangzhou 510530, China. [2] Central Laboratory, School and Hospital of Stomatology, Peking University, Beijing 100871, China. [3] Center for Biomedical Materials and Tissue Engineering, Academy for Advanced Interdisciplinary Studies, Peking University, Beijing 100871, China. [4] CUHK-GIBH Joint Research Laboratory on Stem Cells and Regenerative Medicine, School of Biomedical Sciences, Faculty of Medicine, the Chinese University of Hong Kong, Hong Kong, China. * These authors contributed equally to this work. Correspondence and requests for materials should be addressed to X.S. (email: shu_xiaodong@gibh.ac.cn) or to D.P. (email: pei_duanqing@gibh.ac.cn).

Reprogramming of somatic cells into pluripotent ones with defined factors not only provides a new way to generate functional cells for regenerative medicine, but also establishes a new paradigm for cell fate decisions. For the latter, a cell at a terminally differentiated state can be restored back to pluripotency under well-defined conditions fully observable through molecular and cellular tools. Indeed, the reprogramming process has been analysed in great detail to reveal novel insights into the mechanism of cell fate changes[1–3]. Of particular interest is the acquisition of epithelial characteristics from mesenchymal mouse embryonic fibroblasts (MEFs) commonly employed as starting cells in reprogramming experiments[4]. Termed the mesenchymal to epithelial transition (MET), we and others have described the MET as marking the earliest cellular change upon the simultaneous transduction of reprogramming factors POU5F1 (OCT4), SOX2, KLF4 and MYC or OSKM into MEFs[5,6]. However, when delivered sequentially as OK + M + S, they initiate a sequential epithelial to mesenchymal transition (EMT)-MET process that drives reprogramming more efficiently than the simultaneous approach[7], suggesting that the switching between mesenchymal and epithelial fates underlies the reprogramming process, that is, the acquisition of pluripotency. We then speculated that such a sequential EMT–MET process might underlie cell fate decisions in other situations such as differentiation, generally viewed as the reversal of reprogramming with the loss of pluripotency. Herein, we report that a similar epithelial–mesenchymal–epithelial transition drives the differentiation of human embryonic stem cells (hESCs) towards hepatocytes. A synchronous EMT occurs during the formation of DE and DE cells are in a typical mesenchymal-like status, while further differentiation of DE to hepatocyte-like cells is accompanied by a MET. We reveal that the intermediate mesenchymal DE cells is induced by an autocrine TGFβ signalling and mediated by SNAI1. On the other hand, the neural differentiation of hESCs is not dependent on TGFβ signalling or SNAI1. Thus, EMT-related transcription factor such as SNAI1 participates in lineage-specific cell fate changes.

## Results

### A sequential EMT–MET connects hESCs to hepatocytes.

Human embryonic stem cells robustly express E-cadherin (CDH1) and are epithelial cells in a pluripotent state. Conversely, hepatocytes are also epithelial cells, but are somatic and fully differentiated. Naively it seems possible that epithelial hESCs could move directly to hepatocytes with the gradual loss of pluripotency and gain of hepatic characteristics, without the necessity to pass through a mesenchymal state. To map the cell fate changes along the differentiation pathway between hESCs and hepatocytes, we adopted a serum-free, chemically defined protocol of hepatic differentiation of hESCs based on the stepwise addition of Activin A, FGF4/BMP2, HGF/KGF and then Oncostatin M[8,9]. As shown in Fig. 1a, there were distinct stages marked by POU5F1/NANOG (pluripotency), SOX17/FOXA2 (definitive endoderm, DE), HNF4A/AFP (hepatoblast) and albumin (ALB)/TTR (hepatocyte-like cell) at days 0, 3, 13 and 21, respectively. The cells at day 21 showed typical metabolic activities of hepatocytes such as ALB secretion, synthesis of glycogen or urea, uptake of low-density lipoprotein (LDL) and so on (Supplementary Fig. 1), indicating the effectiveness of the protocol. We characterized the molecular signature of this process first by performing RNA-seq analysis of a time course from days 0 to 21, and compared it with the RNA-seq data of primary human hepatocytes and liver[10–12]. Principal component (PC) analysis indicated that the cells transitioned from pluripotent

stem cell to DE then to hepatocyte-like state (Fig. 1b), based on the gene loading for the respective PCs (Supplementary Fig. 2). In addition, we noticed that PC2 and PC3 contain many EMT-related genes that were dynamically regulated during the hepatic differentiation of hESCs ( Fig. 1c; Supplementary Fig. 2). We next performed real-time RT-polymerase chain reaction (PCR) analysis which confirmed the induction of mesenchymal genes at the DE and hepatoblast stages of hepatic differentiation (Fig. 1d). For example, the mesenchymal gene CDH2, VIM and SNAI1 were all upregulated from D3 to 13 then they were gradually downregulated in the more mature hepatocyte-like cells at D21. The epithelial marker CDH1 showed the opposite expression pattern. Mesenchymal transcriptional factors such as SNAI2, ZEB1 and KLF8 were also dynamically regulated.

We further analysed the expression of these genes at protein level by immunofluorescence staining. As shown in Fig. 2a, CDH1 was clearly downregulated at day 3 and it was re-established in some of the ALB positive cells at day 21. CDH2 was induced at day 3 and maintained up to day 13 in the alpha fetoprotein (AFP) positive hepatoblasts, however, it was clearly downregulated in the ALB positive hepatocyte-like cells (Fig. 2a). Other mesenchymal-related changes such as the expression of VIM and the formation of stress fibres were also present in DE cells. In addition, we measured the migration activity of those cells by scratch assay (Fig. 2b,c) and found that hESCs showed very limited migration ($75 \pm 17 \mu m/24 h$) while cells at day 3 ($395 \pm 15 \mu m/24 h$) or 13 ($505 \pm 14 \mu m/24 h$) were highly motile, further indicating the mesenchymal-like statues of those cells. Cells of D21 had a migration activity of $274 \pm 20 \mu m/24 h$ (Fig. 2c) which was significantly slower than that of D13 ($P = 1.14781E - 15$), suggestive of the cells losing their mesenchymal phenotype. Together, these results indicate that a sequential epithelial–mesenchymal–epithelial transition underlies the differentiation of hESCs to hepatocyte-like cells.

### hESCs begin differentiation with a near synchronous EMT.

The bulk RNA-seq and quantitative PCR (qPCR) analyses revealed a global epithelial–mesenchymal–epithelial transition during the hepatic differentiation of hESCs, but they cannot reveal heterogeneity in the differentiation process. To resolve this process further at the single-cell level, we performed single-cell qPCR with 46 selected genes (Supplementary Table 3) and two control genes (GAPDH, ACTB) on 501 cells (Fig. 3a; Supplementary Data 1) and constructed relational networks of the gene expression of all cells (Fig. 3b). Surprisingly, the pluripotency genes POU5F1 and NANOG were downregulated but not extinguished by day 3. On the other hand, the DE markers SOX17 and GATA6 were robustly upregulated by day 3 and GATA4 and HNF4A are induced slightly later. Upregulation of CDH2 and downregulation of CDH1 were also clearly seen at day 3. Furthermore, cells at day 3 showed substantial homogeneity in their response to Activin A and acquisition of a DE character ($R^2 = 0.74$ compared with the bulk RNA-seq, indicating that the bulk RNA-seq recapitulates the single cell data), suggesting a near synchronous traversal through the EMT. This is remarkable, perhaps reflecting either homogeneous starting hESCs or the synchronization power of Activin A.

We then took advantage of the single-cell qPCR analysis, which when organized into relational maps suggests the temporal order of DE acquisition and EMT, to analyse the possible relationship of EMT and DE formation. When we clustered the single-cell qPCR correlated gene expression for days 0–3 (Fig. 3c), we detected two major clusters centred on either CDH1 or CDH2. The CDH1 cluster contained POU5F1 and SOX2, that is, the pluripotent state, while the CDH2 cluster contained the major DE

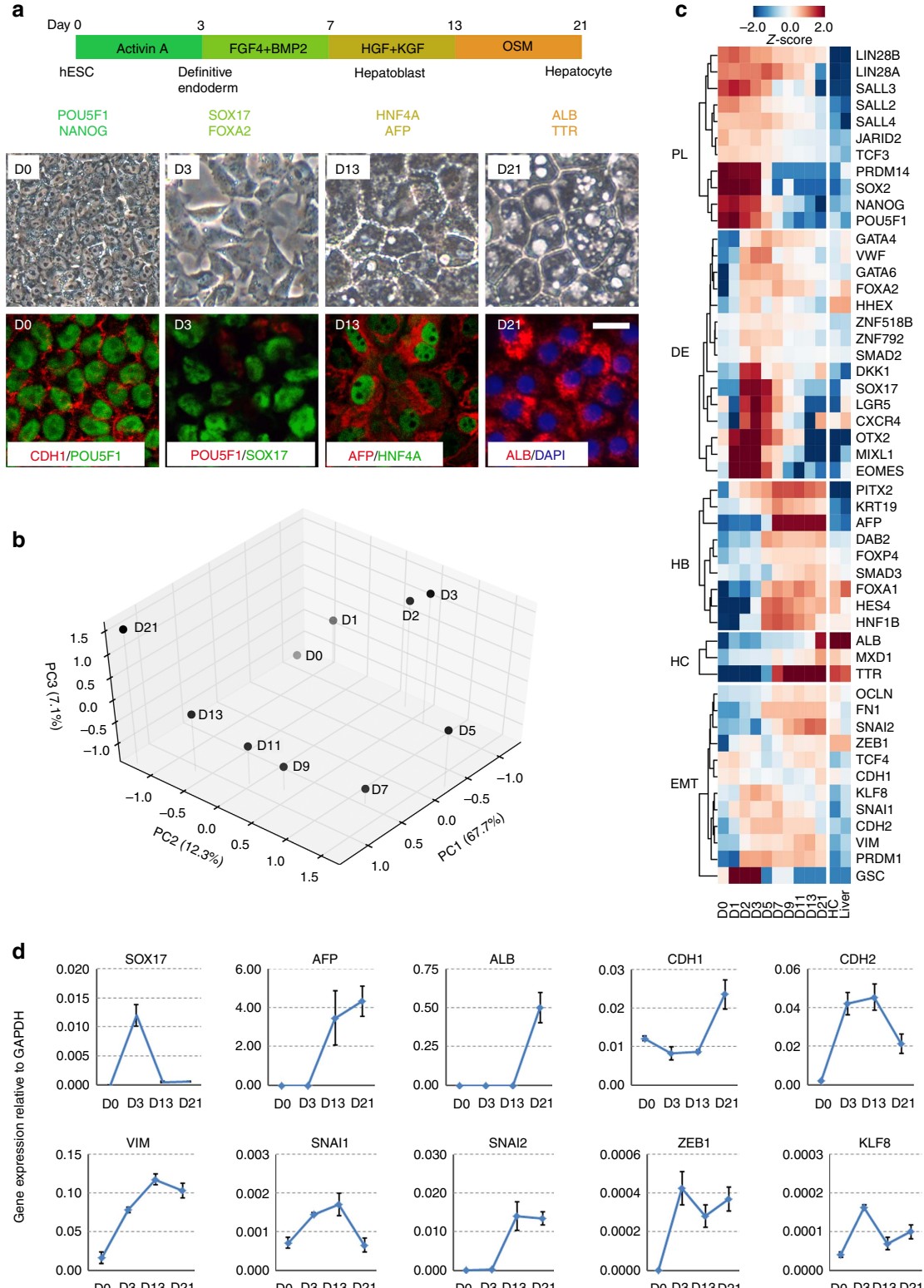

**Figure 1 | Gene expression analysis of the hepatic differentiation of hESCs.** (**a**) Schematic of the protocol and stages of the hepatic differentiation of hESCs. Representative live photos as well as images of immunofluorescence staining of marker molecules at the indicated stages were shown. Scale bar, 20 μm. (**b**) Principal component (PC) analysis indicates the transitions that occur during differentiation. Gene loadings for PC1-3 were listed in Supplementary Fig. 2. (**c**) Expression of selected marker genes from the RNA-seq data. RNA-seq data of primary human hepatocytes (HC) and liver were listed as positive control. From top to bottom, the marker genes are: pluripotency (PL), definitive endoderm (DE), hepatoblast (HB), hepatocyte (HC) and EMT-related genes. RNA-seq data from primary human hepatocytes were taken from GSE43984, GSE57312 and liver SRR002322, GSE13652. (**d**) qRT-qPCR analysis of listed genes at the indicated stages of hepatic differentiation of hESCs (normalized to GAPDH). Data represent mean ± s.d. from three biological repeats.

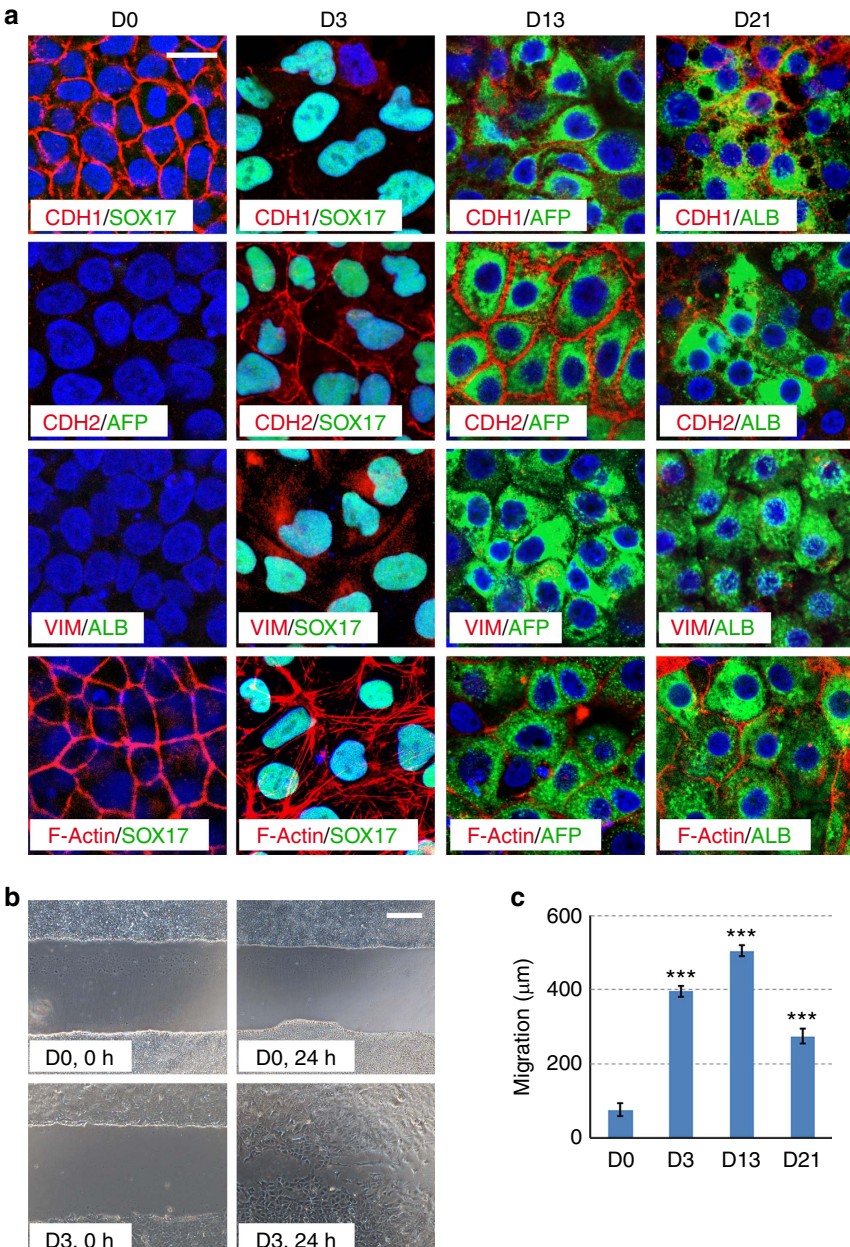

**Figure 2 | An epithelial–mesenchymal–epithelial transition during the hepatic differentiation of hESCs. (a)** Immunofluorescence staining of EMT and cell fate markers at the indicated differentiation stages. Scale bar, 20 μm. **(b,c)** Migration assays for hESC and DE cells. Scale bar, 100 μm. (Data represent mean ± s.d. from three technical repeats. ***$P < 0.001$ when compared to D0 using Student's unpaired $t$-test).

marker genes *FOXA2*, *GATA6*, *GATA4* and *SOX17*. To map the precise timing in more detail we generated scatter plots of individual cell expression of *CDH1* or *CDH2* against the DE marker genes *SOX17* and *GATA6* (Fig. 3d). We showed that *SOX17* and *GATA6* expression was mutually exclusive with *CDH1* expression at day 3, whilst conversely *SOX17* and *GATA6* were both coincident with *CDH2*. These results revealed the occurrence of EMT in all DE cells at the single cell resolution and implied a possible role of EMT during the conversion of hESCs to DE cell fate.

**TGFβ induced by Activin A drives EMT and DE formation.** While a similar EMT was first noticed during the Activin A-induced DE formation of hESCs in a low-serum differentiation media[13], it is currently unclear what signal induces this EMT and whether or not the EMT plays a functional role during the

acquisition of a DE cell fate. Our serum-free and chemically defined differentiation protocol provided an ideal model to investigate these questions. Activin A is not a strong inducer of an EMT by itself so it might function through the induction of other EMT-inducing signals. TGFβ1 is one of the best characterized inducer of EMT and, although no exogenous TGFβ1 protein was used in our differentiation system, we noticed from our RNA-seq data that TGFβ1 mRNA was strongly induced by Activin A. We confirmed the strong induction of *TGFβ1* gene after Activin A treatment by qRT-PCR (Fig. 4a). Indeed, we detected physiological levels of TGFβ1 protein ($1.5 \, \text{ng ml}^{-1}$) in the conditioned medium at day 3 by enzyme-linked immunosorbent assay (ELISA) (Fig. 4b). To test whether the endogenously produced TGFβ1 is involved in the EMT and DE formation, we blocked the TGFβ signalling by a small chemical inhibitor Repsox[14] and determined its effect on EMT and DE

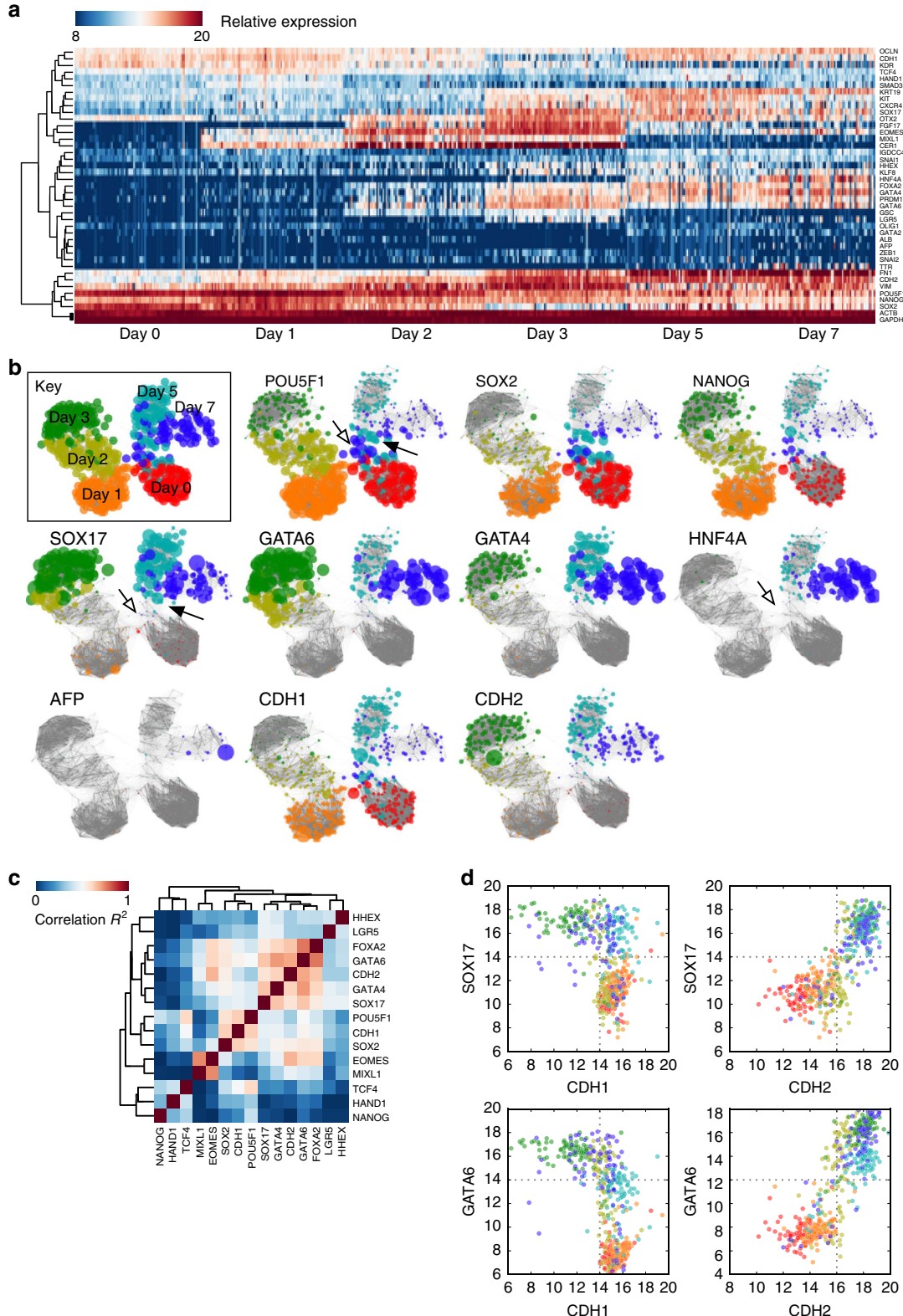

**Figure 3 | Single cell qPCR analysis reveals a synchronous EMT during the differentiation of hESCs to DE.** (**a**) Heatmap of the expressions of selected genes. A complete list of genes used in the study can be found at Supplementary Table 3. (**b**) Relational network plots. Different colours indicate specific days of treatment, node sizes are 2^(relative expression). Open arrows indicate a population of day 5/7 hESC-like cells expressing *POU5F1*, *SOX2*, *NANOG*. Closed arrows indicate a population of cells simultaneously expressing the pluripotent marker genes *POU5F1*, *SOX2*, *NANOG* and the DE markers *SOX17*, *GATA4* and *GATA6*. (**c**) Correlation of gene expression for the indicated genes for days 0 through 3. (**d**) Scatter plots of single cell gene expression for *SOX17* and *GATA6* versus the epithelial marker gene *CDH1* and the mesenchymal marker gene *CDH2*. Colours are the same as in **b**.

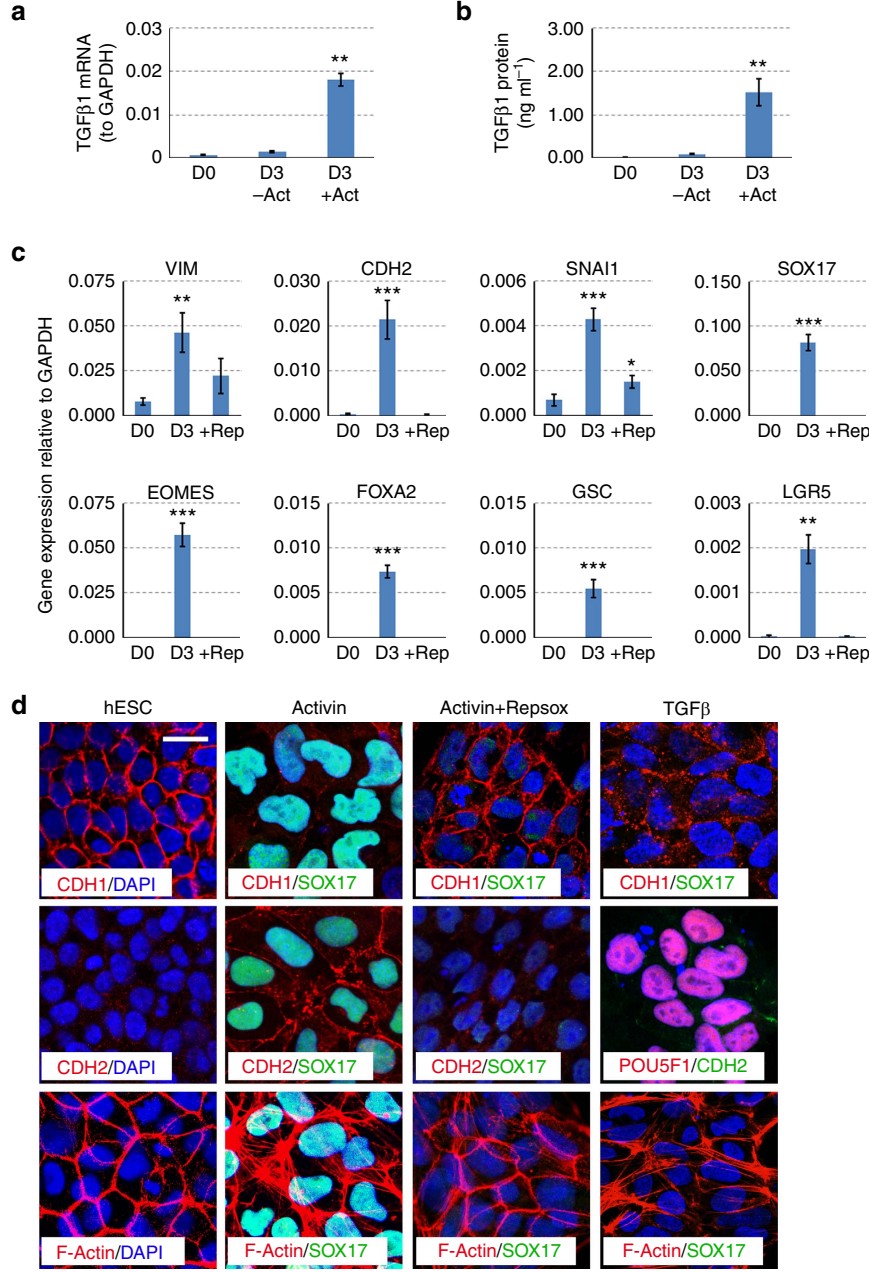

**Figure 4 | TGFβ mediates Activin A-induced EMT and DE formation.** (**a**) qRT-PCR analysis of TGFβ1 expression levels in un-induced hESCs, or in differentiation media with or without Activin A (Act). The expression level of *GAPDH* is arbitrary set as 1. Data represent mean ± s.d. from three independent biological repeats (**P < 0.01 when compared to D0 using Student's unpaired *t*-test). (**b**) Secreted protein levels of TGFβ in conditioned medium at day 3 detected by ELISA. Data represent mean ± s.d. from three independent repeats (**P < 0.01 when compared to D0 using Student's unpaired *t*-test). (**c**) qRT-PCR analysis for the indicated genes in the absence or presence of Repsox (Rep). Data represent mean ± s.d. from three independent repeats (*P < 0.05; **P < 0.01; ***P < 0.001 when compared to D0 using Student's unpaired *t*-test). (**d**) Immunofluorescence staining of Activin A-induced cells with or without Repsox. The right panels are cells treated with TGFβ instead of Activin A. Scale bar, 20 μm.

formation. As shown in Fig. 4c, the induction of mesenchymal markers such as *VIM*, *CDH2* and *SNAI1* were blocked by Repsox. Furthermore, other EMT changes such as the downregulation of CDH1 and formation of F-Actin stress fibre were also inhibited by Repsox (Fig. 4d). We then measured the expression of mesoendoderm markers such as *EOMES*, *GSC*, *SOX17*, *FOXA2* and *LGR5* and found that they all failed to be induced in the presence of Repsox, and these cells also die in hepatic specification media (BMP2 + FGF4) thus failing to be induced into hepatoblasts. These results indicated that the Activin A-induced autocrine/paracrine of TGFβ is not only essential for

the EMT but also plays an indispensable role in the acquisition of DE cell fate. We then determined whether TGFβ is sufficient to induce DE formation in the differentiation media in the absence of Activin A. As shown in Fig. 4d, we found that TGFβ was able to activate part of the EMT programme (downregulation of CDH1, formation of F-Actin) but it failed to downregulate the pluripotency gene (POU5F1) and was not able to stimulate the expression of SOX17.

**SNAI1 is the key EMT factor required for DE formation.** To further investigate the requirement for TGFβ signalling in

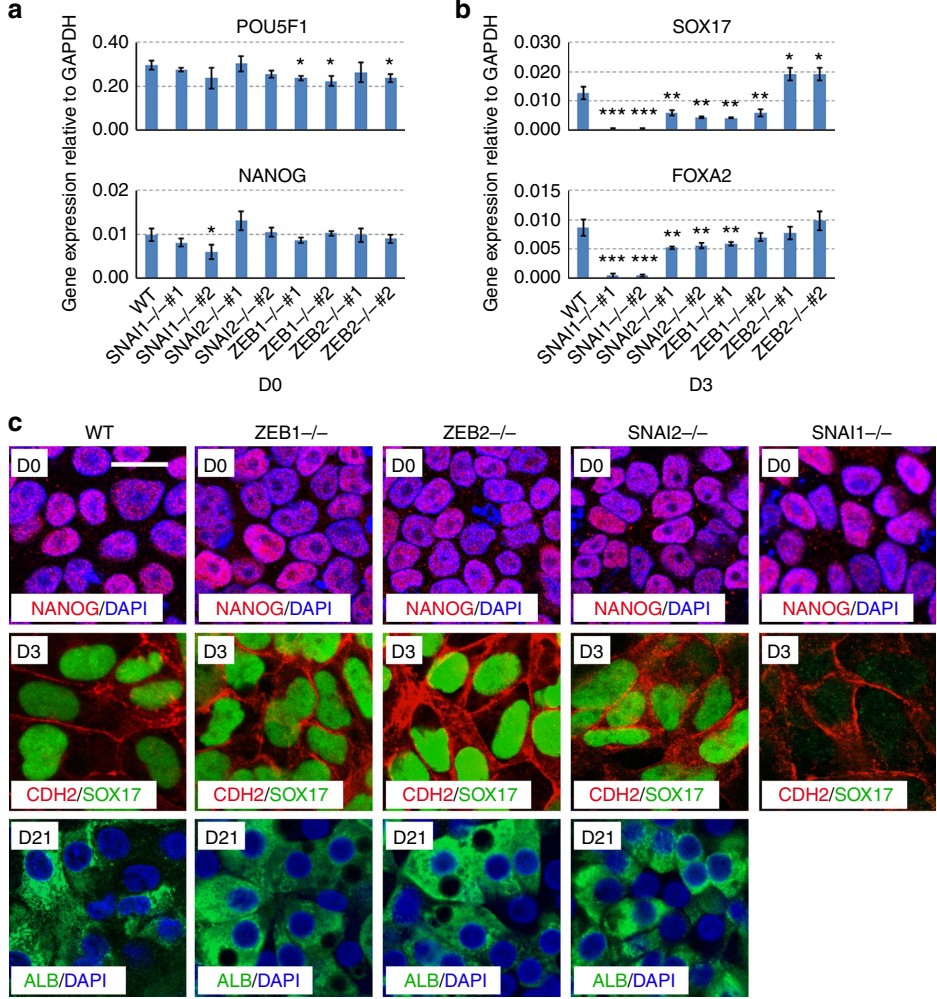

**Figure 5 | Analysis of the role of EMT-related transcriptional factors in hepatic differentiation of hESCs.** (**a**) qRT-PCR analysis of the expression of pluripotency genes in wild-type (WT) and *SNAI1/2* and *ZEB1/2* knockout cell lines. Significance between WT and mutants were determined using Student's unpaired *t*-test (*P < 0.05). (**b**) qRT-PCR analysis of the expression of DE markers in WT and mutant hESC lines. Data represent mean ± s.d. from three biological repeats in **a,b**. Significance between WT and mutants were determined using Student's unpaired *t*-test (*P < 0.05; **P < 0.01; ***P < 0.001). (**c**) Immunofluorescence staining analyses for the hepatic differentiation of hESC lines. Scale bar, 20 μm.

EMT and DE induction, we first focused on the SNAI and ZEB family mesenchymal transcriptional factors upregulated at day 3 (Fig. 1). We knocked-out *SNAI1/2* and *ZEB1/2* by CRISPR/Cas9-mediated gene targeting then examined their effects on DE formation. The strategy of gene targeting is outlined in Supplementary Fig. 3 and we successfully obtained multiple biallelically targeted cell lines for all of them. The resulting mutant cell lines appeared morphologically normal and the expression of pluripotency genes such as *POU5F1* and *NANOG* was maintained (Fig. 5a), indicating these EMT factors are dispensable for the maintenance of pluripotency. We then tested the ability of Activin A to induce DE differentiation in these cell lines. *SNAI1* knockout severely inhibited the induction of *SOX17* and *FOXA2* (Fig. 5b). *SNAI2* or *ZEB1* deficiency moderately reduced the induction of *SOX17* while *ZEB2* knockout curiously stimulated a small induction of *SOX17*, suggesting some kind of compensation effect (Fig. 5b). We confirmed this observation by immunofluorescence staining (Fig. 5c), which showed that *SNAI1* mutants failed to induce the expression of SOX17. Similar to the Repsox-treated wild-type (WT) cells, *SNAI1* mutants did not survive in the hepatic specification media, thus could not be induced into hepatoblasts. On the other hand, *SNAI2* or *ZEB1/2* mutants were able to be further induced into hepatocyte-like cells

at efficiency similar to that in WT hESCs, indicating that ultimately SNAI2, ZEB1 and ZEB2 are dispensable for hepatocyte differentiation and that SNAI1 is required for EMT and DE induction.

We further characterized the function of SNAI1 in the Activin A-induced EMT and DE formation. As expected, *SNAI1* deficiency prevented the downregulation of *CDH1* both at the mRNA and protein levels by Activin A (Fig. 6a,b). In contrast, *CDH2* induction by Activin A was not affected by *SNAI1* (Figs 5c and 6a). Additional EMT-related changes such as the induction of *VIM* and the formation of F-Actin also occurred in the Activin A-treated cells in both WT and *SNAI1* deficient cells (Fig. 6a,b). We then measured the migration capacity of these cells and showed that *SNAI1* deficient cells migrate poorly compared to the WT cells (Fig. 6c,d). Thus, SNAI1 is responsible for part of the Activin A-induced EMT programme (downregulation of *CDH1*, cell migration). On the other hand, the *SNAI1* deficient cells failed completely to activate mesoendoderm/DE lineage markers such as *GSC*, *EOMES*, *SOX17* and *LGR5* compared to the WT cells (Fig. 6a,b), suggesting that SNAI1 is required for Activin A-induced DE formation.

Our observation that inhibition of TGFβ signalling blocked the Activin A-induced activation of SNAI1 as well as DE formation

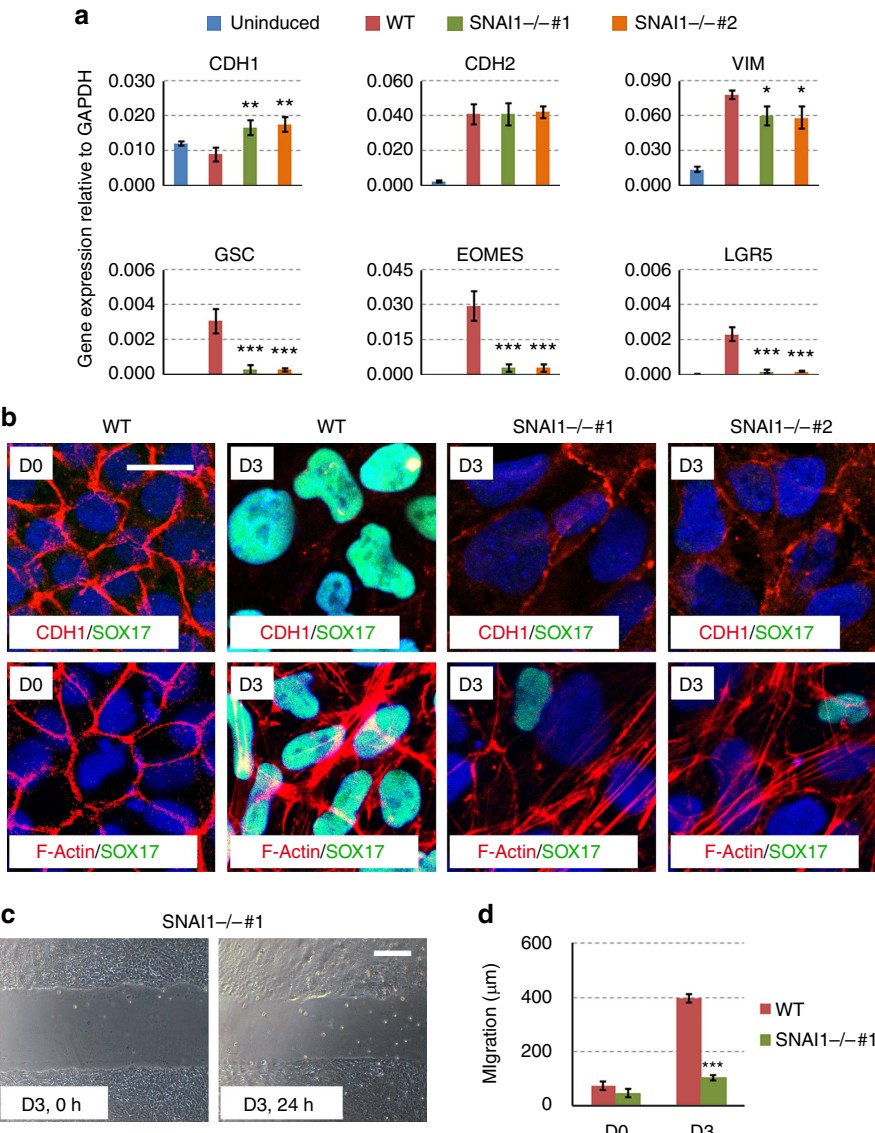

**Figure 6 | SNAI1 mediates Activin A-induced EMT and DE formation.** (**a**) qRT-PCR analysis of EMT and lineage genes in WT and *SNAI1* knockout lines. Data represent mean ± s.d. from three biological repeats. Significance between WT and mutants were determined by Student's unpaired *t*-test (*$P<0.05$; **$P<0.01$; ***$P<0.001$). (**b**) Immunofluorescence staining of WT and *SNAI1* knockout hESCs at D3. Scale bar, 20 μm. (**c,d**) Migration assays for WT and *SNAI1* knockout lines at D3. Scale bar, 100 μm. Significance between WT and mutant was determined by Student's unpaired *t*-test (data represent mean ± s.d. from three technical repeats,***$P<0.001$).

(Fig. 4c), together with the results from *SNAI1* knockout cells, suggested SNAI1 as the major mediator of the TGFβ induced DE formation. So we tested whether overexpression of SNAI1 could rescue the defect in DE formation by Repsox treatment. We established a hESC cell line stably expressing SNAI1 and showed that it maintained the expression of pluripotency gene such as *NANOG* and an epithelial state when maintained in mTeSR1 media (Fig. 7a). We then treated this cell line with Activin A in the presence of Repsox and found that overexpression of SNAI1 promoted the degradation of CDH1 in the presence of Repsox, however, it was not sufficient to restore the expression of DE markers such as SOX17 (Fig. 7b). These results suggest that SNAI1 is required but not sufficient to mediate Activin A-induced DE formation.

**SNAI1 is not required for neural differentiation of hESCs.** We next tested if SNAI1 is involved in the exit of pluripotency or differentiation of hESCs to lineages other than DE. To this end,

we determined the neural differentiation capacity of *SNAI1* deficient hESCs. Neural differentiation was induced by dual inhibition of SMAD signalling (2i, SB431542 + compound C). Neural lineage markers such as *SOX2*, *PAX6* and *OTX2* were induced at similar efficiency in both the WT and *SNAI1* deficient cells at day 5 (Fig. 8a). Meanwhile, pluripotency markers such as *POU5F1* and *NANOG* were clearly downregulated during the same period (Fig. 8a). These results indicated that *SNAI1* deficient cells exit the pluripotency state normally and differentiate into a neural lineage. We examined the EMT-related changes in this process and found that in WT cells CDH1 was downregulated while stress fibre was not obviously formed during neural differentiation (Fig. 8b). Downregulation of CDH1 was less efficient in *SNAI1* mutants which is consistent with the function of SNAI1 in suppressing CDH1during EMT, and this abnormality seemed not to affect the induction of neurectoderm marker gene *PAX6* (Fig. 8b). These results indicated that SNAI1 is not required for neural differentiation of hESCs.

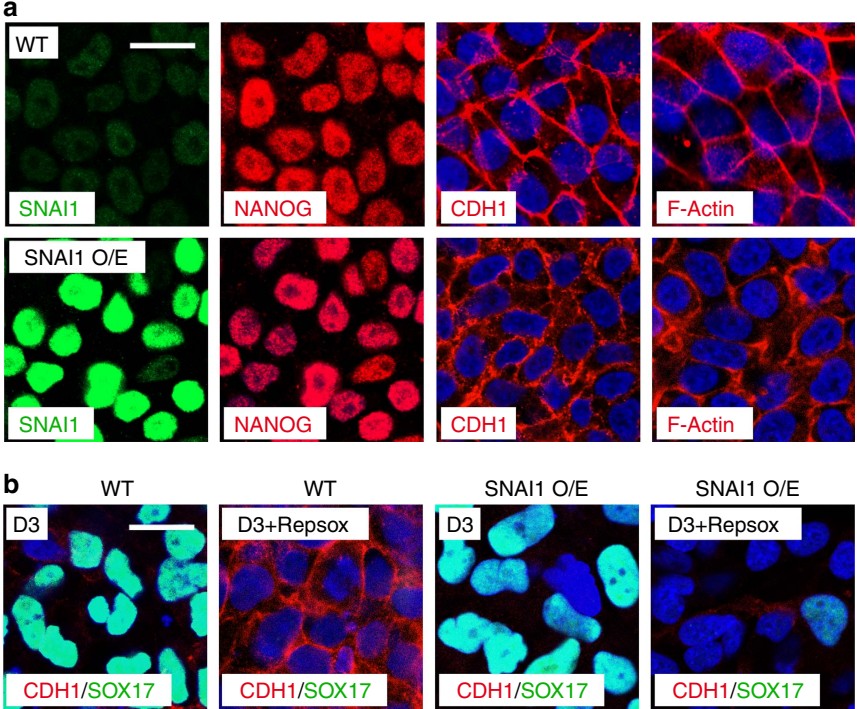

**Figure 7 | Analysis of pluripotency and DE formation in the SNAI1 overexpression hESCs.** (**a**) hESCs overexpressing SNAI1 (SNAI1 O/E) show normal staining pattern for NANOG, CDH1 or F-Actin. Scale bar, 20 μm. (**b**) Immunofluorescence staining of CDH1 and SOX17 after Activin A treatment in the presence or absence of Repsox in WT and SNAI1 overexpression hESCs (SNAI1 O/E). Scale bar, 20 μm.

## Discussion

The transitions between mesenchymal and epithelial states, that is, both the EMT and its reversed process, the MET, have been observed in many processes during normal development as well as in diseased conditions such as metastasis[15–17]. We previously reported that an MET initiates cell fate change during somatic reprogramming of MEF[5] and a sequential EMT–MET process is beneficial for optimal somatic reprogramming[7]. In this study, we presented evidence here that a sequential EMT–MET process drives the hepatic differentiation of hESCs. Together, our studies indicate that an EMT/MET underlies cell fate conversions in both reprogramming and in differentiation along an endoderm cell fate. It is possible that EMT/MET induces cellular reorganization and changes a cell's responsiveness to the extracellular stimuli and/or rewires epigenetic regulatory circuits thus modifies the outcome of a given stimuli. It is interesting that although the first EMT is synchronous and the later MET is asynchronous. We speculate that the lack of a synchronous MET in later stages of *in vitro* differentiation may be related to the lack of full maturity in these induced hepatocyte-like cells. Our discoveries about the functions of EMT/MET in cell fate changes *in vitro* might have *in vivo* implications as well. It is well established that mesendoderm cells undergo an EMT and migration during gastrulation. It is generally thought that migration brings a cell to its destiny where it receives signals required for differentiation. It remains an open question whether or not EMT plays additional roles in cell fate conversions *in vivo*. In a chemically defined *in vitro* differentiation system such as the one we used here, cells are accessible to the induction signal without the need to migrate. Thus, it is an ideal model to investigate the involvement and mechanism of EMT/MET-related changes during lineage-specific differentiations.

Surprisingly, we identified an autocrine TGFβ signalling process during the Activin A-induced EMT and DE formation. We further showed that SNAI1, a master regulator of EMT, is required for the suppression of CDH1 but is dispensable for the induction of CDH2 and F-actin. It remains to be determined whether other EMT-related transcriptional factor, either alone or together with SNAI1, regulates the expression of CDH2 and formation of stress fibres during DE formation. We further showed that the differentiation of DE into hepatocyte involves a MET. By choice, we did not pursue the mechanism associated with this MET process. In that regard, we have noticed that HNF4A is highly induced in the hepatoblast stage and it could be a critical regulator of the subsequent MET. HNF4A is required for the formation of hepatic epithelium and liver architecture in mouse development[18] and is capable of promoting MET and hepatic maturation of hepatoblasts derived from hPSCs[19]. Furthermore, it is an essential factor used in direct conversion of both mouse and human fibroblasts to hepatocytes[20,21]. Together, we propose that HNF4A, together with the downregulation of EMT factor such as SNAI1, plays a critical role in the MET phase of hepatic differentiation.

The *in vivo* function of SNAI1 has been studied in mouse. *Snai1* knockout mice are embryonic lethal with abnormal mesoderm formation, presumably due to the failure to down-regulate CDH1 and initiate EMT during gastrulation[22]. The *Snai1* knockout mouse embryonic stem cells (mESCs) have normal self-renewal but show reduced mesoderm and enhanced neuroectoderm commitment in an embryoid body differentiation system[23]. In the same assay, the induction of SOX17 appears unaffected in the mutant mouse cell lines. However, conversely to the hESC date, ectopic expression of SNAI1 in mESCs induces an EMT (upregulation of CDH2 and downregulation of CDH1) and promotes mesoderm differentiation[24]. These results demonstrate that SNAI1 is not required for self-renewal of mESCs and it favors the mesoderm differentiation during germ layer commitment. Our loss-of-function and gain-of-function studies reveal that in hESCs SNAI1 is dispensable for the maintenance of pluripotency, which is consistent with the mouse data. We found

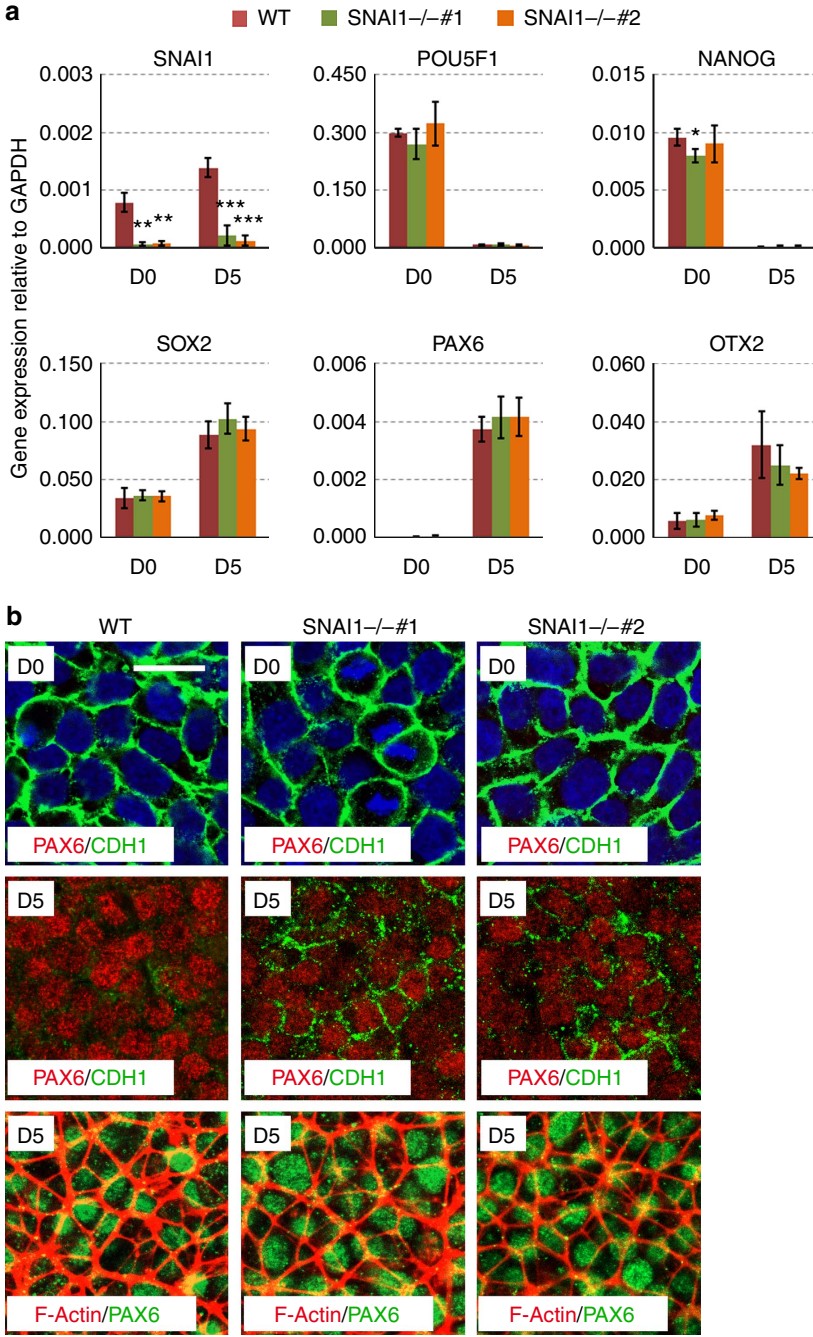

**Figure 8 | SNAI1 is not required for the neural differentiation of hESCs. (a)** qRT-PCR analysis of pluripotency and neural lineage marker in WT and *SNAI1* knockout lines after 5 days of 2i treatment. Data represent mean ± s.d. from three biological repeats. Significance between WT and mutants were determined by Student's unpaired *t*-test (*$P < 0.05$, **$P < 0.01$; ***$P < 0.001$). **(b)** Immunofluorescence staining of WT and *SNAI1* knockout hESCs after 2i treatment. Scale bar, 20 μm.

that SNAI1 has lineage specific function during the differentiation of hESCs: it does not affect neural differentiation of hESCs but markedly blocked the Activin A-induced expression of SOX17. The role of SNAI1 in SOX17 induction is thus different between mESCs and hESCs. This discrepancy could be attributed to the different methods of differentiation (embryoid body vs. targeted differentiation), differences in the nature of mESCs and hESCs (naïve vs. primed), or it might indicate a species-specific function of SNAI1. Further investigations in a chemically defined system will be very helpful to definitively reveal the function of SNAI1 in endodermal differentiation of mESCs.

## Methods

**Maintenance and targeted differentiation of hESCs.** Undifferentiated human H1 ES cells (WiCell) were maintained in monolayer culture on Matrigel (BD Biosciences, 354277) in mTeSR1 medium (Stemcell Technologies, 05850) at 37 °C with 5% $CO_2$. Cells were manually passaged at 1:4 to 1:6 split ratios every 3 to 5 days. For hepatic differentiation, we established a serum-free protocol based on previously described protocols with minor modifications[8,9]. Briefly, cells were cultured for 3 days in RPMI/B27 medium (Insulin minus, Gibco, A18956-01) supplemented with 100 ng ml$^{-1}$ Activin A (Peprotech, 120-14E), followed by 4 days with 20 ng ml$^{-1}$ BMP2 (Peprotech, 120-02) and 30 ng ml$^{-1}$ FGF-4 (Peprotech, 100-31) in RPMI/B27 (complete with Insulin, Gibco, 17504-044) medium, then 6 days with 20 ng ml$^{-1}$ HGF (Peprotech, 100-39) and KGF (Peprotech, 100-19) in RPMI/B27 (complete with Insulin), then 8 days with

20 ng ml$^{-1}$ Oncostatin-M (R&D Systems, 295-OM/CF) in hepatocyte culture media (Lonza, cc-3198) supplemented with SingleQuots (without EGF). For neural differentiation, H1 hESCs were cultured at a high density for 5 days in neural differentiation medium (1:1 of DMEM/F12 (HyClone, SH30023.01) supplemented with 1% N2 (Invitrogen, 17502048) and Neurobasal medium (Gibco, A24775-01) supplemented with 2% B27 (Invitrogen, 17504044) supplemented with 5 μM SB431542 (Selleck, S1067) and 1 μM compound C (Selleck, S7306)). All cell lines used were negative for mycoplasma contamination.

**Immunofluorescence staining.** Cells on glass coverslips were fixed in 4% paraformaldehyde for 30 min, washed with PBS for three times and permeabilized in 0.3% Triton X-100 (Sigma, T8787)/PBS for 30 min. After two brief washes in PBS, cells were blocked in 10% FBS (HyClone, SH30088.03) /PBS for 1 h at room temperature. For actin staining, cells were incubated with 1 unit per ml rhodamine phalloidin (Invitrogen, R415) for 60 min at room temperature. For immunofluorescence staining, samples were then incubated with primary antibody (diluted in 1 × PBS with 0.3% Triton X-100/10% FBS) overnight at 4 °C, washed three times with blocking solution and incubated with a secondary antibody for 1 h at room temperature. The cells were washed and counter stained with DAPI (Sigma, D9542) for 5 min, and then imaged with the Zeiss LSM 710 confocal microscope (Carl Zeiss). Antibodies used in this study are listed in Supplementary Table 1.

**qRT-PCR.** Total RNA was extracted using the Trizol reagent (MRC, TR118) and 2 μg of total RNA was reverse-transcribed using the ReverTra Ace qPCR RT Kit (TOYOBO, FSQ-101). The product (cDNA) was properly diluted and used as PCR template. PCR reactions were performed with the SYBR Premix Ex Taq Kit (TAKARA, RR420A) on the CFX96 Touch Real-Time PCR Detection System (Bio-Rad). GAPDH was used as the internal control. Primers used are listed in Supplementary Table 2.

**RNA-seq and bioinformatics.** Total RNA was collected during the differentiation of hESCs to hepatocyte-like cells on days 0, 1, 2, 3, 5, 7, 9, 11, 13 and 21. Approximately 4 μg of RNA was used to generate sequencing-ready cDNA library with the TruSeq RNA Sample Prep Kit (Illumina, RS-122-2001). DNA fragments (250–300 bp) were recovered from the gel slice using the QIAquick gel extraction kit (QIAGEN, 28704). The concentration of cDNA library was determined with the Qubit dsDNA HS Assay Kit (Invitrogen, Q32851). Samples were sequenced on a MiSeq according to the manufacturer's instructions to an average depth of 2 million sequence tags. Reads were aligned to the ENSEMBL (mm10 v76) transcriptome using Bowtie2 (v2.2.0)[25] and RSEM (v1.2.17)[26], GC-normalized using EDASeq (v2.0.0)[27]. Analysis was performed using glbase[28]. Reads were deposited with GEO under the accession number GSE70741.

**Single-cell qPCR analysis.** Single-cell qPCR was performed using a Fluidigm C1 and BioMark HD as described by the manufacturer. Briefly, a cell suspension of a concentration of 166–250 k ml$^{-1}$ was loaded into a 10–17 μm C$_1$ Single-Cell Auto Prep IFC chamber (Fluidigm, PN100-5479), and cell capture was performed on the Fluidigm C$_1$ System. Both the empty wells and doublet-occupied wells were excluded from further analysis. Upon capture, reverse transcription and cDNA pre-amplification were performed using the Ambion Single Cell-to-CT Kit (Life Technologies, PN 4458237) and C$_1$ Single-Cell Auto Prep Module 2 Kit (Fluidigm, PN100-5519). The pre-amplified products were diluted tenfold prior to analysis with TaqMan Gene Expression Master Mix (Life Technologies, 4369016) and inventoried TaqMan Gene Expression assays (20 ×, Applied Biosystems) in 96.96 dynamic Arrays on a BioMark System (Fluidigm). Inventoried TaqMan primers were used (Supplementary Table 3). Relative expression was calculated as described before[29] except that a Ct value of 25 was used for low expressed genes. Relational network plots (mdsquish) were implemented as part of glbase[28]. Briefly, the normalized Euclidean distance between all cells was measured for singular value decomposed PCs 1, 2, 3, and a network was then constructed using a threshold of 0.92 for weak links (dotted lines) and 0.99 for strong links (solid lines) with a maximum of the 50 best scoring edges per node, the network was then laid out using graphviz 'neato' layout. Node sizes are 2$^{(relative\ expression)}$.

**Cell migration assay.** Scratch assay was used to determine the migration activity of H1-derived cells. Briefly, cells in a confluent monolayer were scratched with a needle to form a cell-free zone into which cells at the edges of the wound can migrate. The denuded area was imaged to measure the boundary of the wound at pre-migration. Images of cell movement were captured at regular intervals within a 24 h period for data analysis.

**ELISA.** The protein level of TGFβ in the Activin A stimulated H1 cell culture media was determined with an ELISA Kit (R&D Systems, DB100B) as described by the manufacturer.

**Gene targeting.** The strategy for gene targeting in H1 cells was outlined in Supplementary Fig. 3a. sgRNAs to human SNAI1, SNAI2, ZEB1 and ZEB2 were designed using the software provided by Dr Feng Zhang's lab at crispr.mit.edu and cloned into the pX330 vector. Donor constructs were prepared by inserting gene specific left and right arms into the donor vector with either a puromysin-resistance (pGK-Puro) or a neomycin-resistance (pGK-Neo) cassette flanked by loxP sites. For gene targeting, H1 cells were digested with Accutase (Sigma) for 8 min at 37 °C then electroporated with linearized donor DNA containing pGK-Puro (2 μg), pGK-Neo (2 μg) and pX330-sgRNA (4 μg). Cells were then plated onto Matrigel (Corning)-coated six-well plates in the presence of Y-27632 (10 μM, Sigma, Y0503) for 1 day. Positive colonies were selected by puromycin (0.5 ng ml$^{-1}$, Gibco, A1113803) plus G418 (50 ng ml$^{-1}$, Gibco, 10131035) in mTeSR1. Colonies were transferred to Matrigel-coated 24-well plates to grow up for several days then passaged using 0.5 mM EDTA onto Matrigel-coated 12-well plates. Biallele SNAI1 knockout colonies were first screened by PCR on genomic DNA with the primer F2 and R2 (Supplementary Fig. 3b) and further confirmed by real-time RT-PCR analysis (Supplementary Fig. 3c). The sequences of sgRNA and PCR primers used are listed in Supplementary Table 4.

**Periodic acid Schiff staining.** Periodic acid Schiff (PAS) staining was performed using the PAS staining kit (Polysciences, 24200-1) according to the manufacturer's instructions.

**LDL uptake.** Cells were washed with PBS and incubated in culture medium containing 4 μg ml$^{-1}$ LDL (Invitrogen, L23380) for 30 min at 37 °C. Cells were then fixed with 4% formaldehyde and stained with DAPI (Sigma, D9542). LDL uptake by cells was examined under a fluorescence microscope.

**Indocyanine green uptake and release.** Indocyanine green (ICG) (Sigma, 1340009) was dissolved in DMSO at 5 mg ml$^{-1}$. When cells were ready, ICG was diluted freshly in culture medium to 1 mg ml$^{-1}$. Diluted ICG was added to cultured cells for 30 min at 37 °C. After washing with PBS, the cellular uptake of ICG was examined under a microscope. Then cells were refilled with the culture medium and incubated for 6 h and the release of cellular ICG was examined.

**ALB and urea secretion.** ALB and urea secretion of differentiated hepatocytes were analysed using fully automatic chemistry analyzer (SHIMADZU CL-8000).

**Statistical analyses.** In general, experiments were done from three biological repeats when possible. Data were presented as mean ± s.d. calculated using Microsoft Excel. Statistical differences were determined by unpaired two-tailed Student's t-test. P-values < 0.05 were considered statistically significant. No statistical method was used to pre-determine sample size. No samples were excluded for any analysis.

**Data availability.** The RNA-Seq data discussed in this publication have been deposited with GEO under the accession number GSE70741. All other relevant data are available from the corresponding authors upon request.

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

## Acknowledgements

The work was supported by Grants from the National Natural Science Foundation of China (31421004, 31530038, 31601100), Ministry of Science and Technology 973 Program (2015CB964700), Strategic Priority Research Program of the CAS (XDA01020307, XDA01020401), Key Research Program of Frontier Sciences, CAS, (QYZDJ-SSW-SMC009), MFPRC (ZDYZ2012-3), Natural Science Foundation of Guangdong Province (2016A030310121) and Science and Technology Planning Project of Guangdong Province (2014B020225002, 2015B020228003, 2016B030301007).

## Author contributions

X. Shu. and D.P. conceived the project, designed experiments and analysed data; Q.L. and S.L. performed single cell analysis, *SNAI1* overexpression in H1 and inhibitor experiments; Y.C. and D.Z. characterized the hepatic differentiation process; Y.S. and B.L. performed *SNAI1/2* and *ZEB1/2* KO in H1; X. Shi. and G.P. helped perform the RNA-seq; A.P.H. performed all bioinformatics; Y.L., W.-Y.C. and S.W. provided reagents and facilitated experiments; Q.L., A.P.H., X. Shu. and D.P. wrote the manuscript.

## Additional information

**Competing interests:** The authors declare no competing financial interests.

