## [Peer Review File · Nature Communications]

Reviewers' Comments:

Reviewer #1 (Remarks to the Author)

In this study, the authors show that during hESC differentiation to hepatocytes the cells first go through a EMT process that is then followed by a MET process to form epithelial cells. The authors suggest that TGFB1 ligand is endogenously induced by Activin A exposure which leads to the induction of SNAI1 that drives the initial EMT process. Although the notion that ESCs need to undergo EMT and then MET during their differentiation toward hepatocytes is not surprising or new, (i.e., during development the definitive endoderm cells must express mesenchymal markers in order to migrate along the developing embryo- this property is not relevant for hepatocyte differentiation but rather to embryonic development), the analysis at the single cell level is exciting and show interesting variation in the late MET process. However, there are several concerns that should be addressed prior to publication.

Major comments

1. Figure 1, the authors should add primary hepatocytes to their analyses. The reviewer would like to know that their differentiated hepatocytes are resemble to endogenous hepatocytes in all examined parameters (staining, gene expression, etc...), especially given the fact that their 21d hepatocytes express many mesenchymal genes too (ZEB1, VIM, FN1, SNAI2, Figure 1C). With this line of thinking, how one can defines an epithelial phenotype or mesenchymal phenotype if the cells express both markers (OCLN, CDH1 together with VIM, FN1, SNAI2).

2. Figures 2F and 4H, the authors should test the migratory potential of the cells with other assays (e.g., transwell)

3. The authors claim "To further resolve the transitions, we then performed immunofluorescence analysis and showed that SOX17 positive cells were all negative for CDH1, but positive for CDH2 (Fig. 3f). More importantly, there were CDH2-positive but SOX17-negative cells, suggesting that EMT has occurred but SOX17 has not yet been sufficiently induced in these cells (Fig. 3f)". In Figure 3F, it seems to the reviewer that there are some CDH2-negative-Sox17 positive cells, if this indeed the case, it is not clear whether EMT precedes the formation of SOX17-positive cells or whether it is a stochastic process. The authors should examine this notion more thoroughly.

4. Figure 4, the authors claim that EMT is induced by the accumulation of endogenous TGFB1 following Activin A exposure. If this is true, the authors should show that the addition of exogenous TGFB1 can replace Activin A and induces at least EMT if not a complete DE differentiation.

5. To test whether the activation of SNAI1 is via the smad3/2/4 complex, the authors should use SIS3 inhibitor that inhibits Smad3 phosphorylation instead of RepSox. In addition, the authors should test the levels of Smad3 phosphorylation following RepSox treatment by western blot.

6. To test whether SNAI1 mediates the effect of TGFB1, the authors should show that SNAI1 overexpression can rescue the negative effect of RepSox.

7. The authors should discuss whether hepatocyte differentiation from ESCs indeed requires EMT-MET process or whether ESC differentiation just mimicking embryo development and DE cells must migrate along the developing embryo to locate the progenitor cells in the correct location in the embryo.

Reviewer #2 (Remarks to the Author)

In the present ms., the authors investigate transient EMT-MET processes in cell fate decisions, in particular during differentiation of hepatocytes from hESCs. The study is based on previous data from the authors and other other groups showing that MET is required for efficient reprogramming of MEFs (Liu et al, 2010; Samavarchi-Tehrani et al., 2010). Also, and of particular interest for the present study is the previous report of the authors showing that sequential delivery of OKMS factors to MEFs (and other mouse cell lines) induce a sequential EMT-MET process driving reprogramming more efficiently than simultaneous addition of the four factors (Liu et al., 2013).

Now, the authors hypothesize that similar sequential EMT-MET process might occur during cell fate decisions such as in hepatocyte differentiation. They approach this interesting question by mapping expression patterns during hepatocyte differentiation from hESCs by bulk RNA-seq. Based in this analysis they observed a synchronous EMT followed by a less synchronous MET. They then perform single cell analyses to confirm the bulk RNA-seq data and further explore the action of TGF β and the requirement of SNAI1 for EMT at initial differentiation stages. The authors provide convincing evidence for early and synchronous EMT process mediated by TGF β at early stages of hESCs differentiation (i.e., Definitive Endoderm), apparently through SNAI1. However, the study fails sort in providing robust evidence for the subsequent MET process proposed to occur for hepatoblast and hepatocyte differentiation. In addition, the study lacks robust statistical analyses required in order to support strong conclusions. On the other hand, the results showing here are a little bit striking when compared to the previous data on EMT-MET during iPS reprogramming, as in both distinct and apparently opposite biological process (reprogramming vs differentiation) the spatio-temporal sequence of events are similar: i.e., EMT followed by MET. It is also unclear if the present data can be extended to other cell fate scenarios distinct from hepatocyte differentiation. Therefore, although the present study provides data of interest to a broad scientific community, further studies are required to provide more convincing evidence to fully support that a sequential EMT-MET process occurs in cell fate decisions during differentiation.

Main points.

1. The authors present evidence for an EMT process occurring at initial stages of differentiation associated to definitive endoderm (from D0 to D3) based on gene expression and marker analyses. They also claim that a subsequent, although asynchronous, MET process occurs concomitant to hepatoblast and hepatocyte differentiation (from D7 to D13-D21) (Figs. 1 and 2). Detection of the MET process is mainly based in re-expression of CDH1 and some other epithelial markers (OCLN) and downregulation of CDH2 and SNAI1. However, a close inspection of gene expression in bulk RNA analyses (Fig. 1c) indicates that expression of many mesenchymal genes (VIM, FN1) and EMT-TFs (ZEB1, SNAI2, TCF4) remain at high level at D13-D21 suggesting that not a true MET process is taking place during differentiation. In my opinion, this is an important point that the authors should further investigate in order to make more strong conclusions on the relevance of the proposed MET process. Apart from expression of hepatocyte markers the authors do not provide strong evidence for the functionality of hepatoblast and hepatocytes obtained at D13 and D21, respectively. Further characterization of the protein expression level and, importantly, localization of the above mentioned mesenchymal genes and EMT-TFs should be performed.
2. In direct relation to the above point, single cell RNA expression analyses also broadly confirm EMT at early points (Fig. 3a). Indeed, downregulation of CDH1 and upregulation of CDH2, and other mesenchymal genes, is clearly detected in individual cells at D3. However, while upregulation of CDH1 starts to be detected at D5 it coexists with apparently high expression levels of CDH2 and other mesenchymal genes up to D7, indicating intermediate mesenchymal/epithelial states. The authors should at least discuss on this important data.
3. Strikingly, upregulation of SNAI1 at D3 at single cell level is hard to detect in the heatmap shown in Fig. 3a. A similar situation occurs for other EMT-TFs, ZEB1 and SNAI2, making difficult to link the apparent EMT to expression of EMT-TFs likely because of their low expression levels. To clarify this relevant issue, relational network and scatter plots correlating the expression of CDH1 and CDH2 with SNAI1, and other EMT-TFs, should be included as those presented for CDH1, CDH2 and with DE selected markers (Fig. 3b-d).
4. The authors provide evidence that TGF β mediates the initial EMT (Fig. 4) apparently through induction of SNAI1. To provide evidence of Snai1 involvement in initial EMT they perform knockdown assays with siRNAs to different EMT-TFs (Fig. 5a, b), claiming that SNAI1 is the most critical EMT-TF for upregulation of CDH2 as well as of DE markers (Fig. 5b). They further show that

SNAI1 knockdown blocks EMT and mesoderm/endoderm marker expression (Fig. 5c). However, the data presented in Fig. 5 lack rigorous statistical analysis that is absolutely required to support this important conclusion. In addition, the effect of SNAI1 in suppressing EMT (i.e., upregulation of CDH1, downregulation of CDH2) needs to be confirmed at the protein and phenotypic level. IF analysis should be performed to confirm the data obtained by qPCR analyses. In addition, migration assays similar to those presented in Fig. 2f, g, should be performed in the presence of SNAI1-siRNA and/or RepSox. These studies will provide more convincing evidence for the requirement of SNAI1 in the initial EMT process mediated by TGF β .

5. The implication of EMT-TFs (SNAI2, TWIST1/2, ZEB1) in the initial EMT, mediated by TGF β , is discarded by the authors based in the apparently low effect of the corresponding siRNAs to repress CDH2 and DE marker induction (Fig. 5 a, b). However, as mentioned above these data lack robust statistical analyses to fully discard the effect of those siRNAs. Indeed, upregulation of SNAI2 and ZEB1 at D3 occurs at similar levels than SNAI1 and all three factors are inhibited to a similar extent by the TGF β inhibitor RepSox, as detected in bulk qPCR analyses (Fig. 4f). These data, in principle, support the implication of at least SNAI2 and ZEB2 in early EMT and need to be further studied by more extended knockdown assays.

6. In the same line, and related to comments on point 1, the potential role of SNAI2 and ZEB1 in late states of differentiation should also be tested. Is there any effect of knocking down these factors in the commitment to differentiation states (D13, D21)?

7. It is also striking that apparently similar spatio-temporal EMT-MET process occurs during cell fate (i.e., hepatocyte differentiation) and during reprogramming, as previously shown by the authors and others. It is also unclear if this situation is specific to hepatocyte differentiation or can be extended to other cell fate process. The authors should at least discuss on this relevant issue and present arguments on this apparent paradox.

8. Statistical analyses need to be included in all relevant Figures (Fig. 2a-c, g; Fig. 4f, h; Fig. 5a-c)

Minor points.

1. Mention to Suppl. Fig S1 is lacking in the text.

2. The different gene categories indicated in the legend to Fig. 1c should be labeled in the Figure panel to ease its reading to non-specialists in hepatocyte or EMT fields.

Reviewer #3 (Remarks to the Author)

In the present study Duanqing Pei and colleagues report evidence for a sequential EMT-MET process during hESC differentiation towards hepatocytes. As an initial experiment they performed in vitro differentiation of human ESCs into hepatocytes and performed bulk RNA-seq experiments throughout the whole differentiation time course. To further analyze heterogeneity of the differentiating cell population they conducted single cell qPCR analysis on 46 relevant genes for early time points of the differentiation process. Some of the findings were further investigated by immunohistochemistry and siRNA mediated knockdown experiments.

The observation of a sequential EMT-MET process during differentiation of pluripotent stem cells into mature epithelial cell types as a general phenomenon would be very interesting and relevant to a broader audience. However, in the current study only a specific system is analyzed and the generality of the finding is unclear. To prove the generality at least one more differentiation system should be analyzed.

Moreover, although the study provides in my opinion clear evidence for an initial epithelial-to-mesenchymal transition, the subsequent mesenchymal-to-epithelial transition is not apparent from the data (see major concerns below). Therefore, the major conclusion of the study is not sufficiently supported by the data presented in the manuscript, and additional experiments and data analysis are required.

Major concerns:

1. Based on the PCA in Figure 1b it is claimed that starting from an epithelial state cells traverse through a mesenchymal state and then revert to an epithelial state. The authors make the statement that PC3 separates an epithelial from a mesenchymal phenotype. However, neither do the authors explain how this was inferred, nor do they show any supporting data. I assume this observation is based on the loadings of epithelial and mesenchymal markers, but this should be reported in the text.
2. Figure 1c and 1d show the down-regulation of epithelial genes and the up-regulation of mesenchymal genes between day 0 and day 3, it appears that mesenchymal genes such as Vimentin remain up-regulated at later timepoints. How can this be reconciled with a subsequent MET during these later timepoints?
3. Figure 2c clearly shows up-regulation of mesenchymal genes at day 3 after normalizing to GAPDH. How highly are these genes expressed at later timepoints (day 13) after normalization to GAPDH?
4. A possible explanation for the remaining high levels of mesenchymal markers could be post-transcriptional down-regulation. In Figure 2d it is shown that no CDH2 protein is detected although it is still up-regulated on the mRNA level (Figure 2s). The authors should measure protein expression at day 13 for additional mesenchymal markers to test this hypothesis.
5. In Figure 2f and g a migration assay is presented, proving the mesenchymal state of DE cells. These experiments should also be performed at later differentiation stages to prove a loss of the mesenchymal character.
6. The single cell gene expression analysis nicely demonstrates that the cell-population is fairly synchronized during the first seven days of the time course. This also shows that the remaining expression of mesenchymal markers is not due to population heterogeneity. The data shown in Figure 3b and c confirm the correlated expression of CDH1 and CDH2 at day 5 and 7. The correlation between CDH1 and other mesenchymal markers should also be analyzed. More importantly, the authors should add data for day 13 or 21 to measure the expression of mesenchymal markers at a later stage.
7. Based on the presence of CDH2-positive cells that do not express SOX17 and the absence of SOX17 positive cells that are negative for CDH2 the authors claim that EMT precedes specification of the DE. These data are not sufficient to infer a temporal order of the events. Perhaps the SOX17-negative cells are able to undergo EMT but unable to differentiate (corresponding to the POU5F1 positive cells at day 7 shown in Figure 3b). This could be tested by co-staining pluripotency markers (see also 8.).
8. The authors perform knockdown experiments of transcription factors involved in the EMT. Based on the data presented in Figure 5c it is apparent that knockdown of SNAI1 has a strong effect on DE specification, but only a moderate effect on the expression of CDH1 and CDH2. How does the expression of other epithelial and mesenchymal markers change? The authors should perform additional experiments (protein staining, migration assay) to demonstrate that the knockdown

inhibits EMT and that this is the primary cause of failure of DE specification. However, even with these data additional experiments are required to support the claim that the EMT precedes DE specification. At the very least it should be tested if EMT still occurs in the absence of DE specification genes by knockdown experiments.

Minor concerns:

1. Figure 2b is not cited in the main text and Figure 2d is discussed prior to Figure 2c.
2. In Figure 3b the authors show relational network plots without discussing how these were derived. What does proximity of two datapoints in this representation reflect and what does a link between two datapoints represent?
3. In Figure 3d the gene names should also be shown along the horizontal axis to ease reading of the heatmap.

Reviewer #4 (Remarks to the Author)

Reprogramming pluripotent cells into hepatocytes is a topic of interest in the field of cell transplantation therapy and developmental biology. The manuscript investigated the EMT-MET processes of hES cells using single cell analysis. The study also identified SNAI1 as a downstream mediator of Activin A-induced differentiation. However, the manuscript is currently unacceptable due to the following reasons:

(1) Statistics: authors did not provide any information about (a) statistical significance, for their entire analyses; (b) number of replicates (biological and/or analytical), for RNA-seq analyses and single cell analyses. Therefore, it was impossible to determine whether the paper is technically sound, and whether the results are reliable.

(2) Novelty: the fact that hES cells treated with Activin A undergo an EMT while differentiating into definitive endoderm has been already reported in the reference no. 11 :

D'Amour KA et al. Efficient differentiation of human embryonic stem cells to definitive endoderm. *Nat Biotechnol.* 2005 Dec;23(12):1534-41.

Authors are claiming that the present study is investigating the acquisition of a genuine mesenchymal phenotype (page 5). To support their claims, additional criteria for EMT-MET such as redistribution of ZO1 and other cytoskeletal proteins at multiple time points including day 21 need to be investigated.

(3) Conclusions: Data presented are not sufficient to support their conclusion that cells transition into a mesenchymal state at day 3 and then revert back to an epithelial state (page 4):

(a) Figure 2: to prove that cells at day 3 indeed acquired mesenchymal characteristics, authors should include appropriate controls such as mesenchymal cells (such as MEFs) and epithelial cells (such primary hepatocytes) in there analyses.

(b) Figures 2F, 2G: it is unclear whether cells lose their migratory phenotype at later time points.

(c) Expression of some of mesenchymal markers is sustained throughout the differentiation process until day 21 (Figures 1C, 1D)

Reviewers' comments:

Reviewer #1 (Remarks to the Author):

In this study, the authors show that during hESC differentiation to hepatocytes the cells first go through a EMT process that is then followed by a MET process to form epithelial cells. The authors suggest that TGFB1 ligand is endogenously induced by Activin A exposure which leads to the induction of SNAI1 that drives the initial EMT process. Although the notion that ESCs need to undergo EMT and then MET during their differentiation toward hepatocytes is not surprising or new, (i.e., during development the definitive endoderm cells must express mesenchymal markers in order to migrate along the developing embryo- this property is not relevant for hepatocyte differentiation but rather to embryonic development), the analysis at the single cell level is exciting and show interesting variation in the late MET process. However, there are several concerns that should be addressed prior to publication.

Response: We appreciate the positive comments about our work. We have now addressed the concerns raised below.

Major comments

1. Figure 1, the authors should add primary hepatocytes to their analyses. The reviewer would like to know that their differentiated hepatocytes are resemble to endogenous hepatocytes in all examined parameters (staining, gene expression, etc...), especially given the fact that their 21d hepatocytes express many mesenchymal genes too (ZEB1, VIM, FN1, SNAI2, Figure 1C). With this line of thinking, how one can defines an epithelial phenotype or mesenchymal phenotype if the cells express both markers (OCLN, CDH1 together with VIM, FN1, SNAI2).

Response: we have added RNA-seq hepatocytes and liver data (from GSE ERR030895, SRR002322, and SRR014264) and data from primary human hepatocytes (SRR656273 and SRR1266980). This indicates that many lineage markers in our day 21 sample shows similar expression to primary liver tissue and hepatocytes (Fig. 1c) . However, this does not mean that we claim our cells are fully mature in vitro equivalents of human liver. Indeed, our D21 cells still have some way to go before they fully resemble in vivo hepatocytes (such as further downregulation of HB and EMT related genes). Note that for this PCA and all PCA in the manuscript we use a new implementation (sklearn, versus scipy in the previous manuscript) that centers and whitens the raw data, and consequently the informative PCs now begin at PC1. No conclusions were altered based upon this change.

In addition, in vitro differentiation products are usually heterogeneous cell mixture of various cell types and differentiation stages. So we have immunostained D21 cells to show that the ALB-positive cells were negative for many of the mesenchymal markers mentioned above (Figure 2a and the enclosed picture below).

2. Figures 2F and 4H, the authors should test the migratory potential of the cells with other assays (e.g., transwell)

Response: We attempted to perform a transwell assay, as the reviewer suggests, but it did not work well due to technique problem. The use of transwell seems not desirable for the attachment, growth and differentiation of hPSCs. The migration assay (Fig. 2b, c) used here is suitable for assessing motility in this instance.

3. The authors claim "To further resolve the transitions, we then performed immunofluorescence analysis and showed that SOX17 positive cells were all negative for CDH1, but positive for CDH2 (Fig. 3f). More importantly, there were CDH2-positive but SOX17-negative cells, suggesting that EMT has occurred but SOX17 has not yet been sufficiently induced in these cells (Fig. 3f)". In Figure 3F, it seems to the reviewer that there are some CDH2-negative-Sox17 positive cells, if this indeed the case, it is not clear whether EMT precedes the formation of SOX17-positive cells or whether it is a stochastic process. The authors should examine this notion more thoroughly.

Response: As suggested, we have explored this angle. We first performed immunostaining that indicate that cytoskeletal changes occur before cells become SOX17 positive (see the enclosed figure below).

Further, we performed siRNA mediated knockdown of SOX17 and found that depletion of SOX17 effectively blocks the induction of DE markers such as FOXA2, but it does not affect EMT related changes (F-actin formation, induction of CDH2, SNAI1) (see the enclosed figure below). Together, these observations indicate that EMT occurs prior to Sox17 expression as we previously stated.

4. Figure 4, the authors claim that EMT is induced by the accumulation of endogenous TGFB1 following Activin A exposure. If this is true, the authors should show that the addition of exogenous TGFB1 can replace Activin A and induces at least EMT if not a complete DE differentiation.

Response: In order to identify whether Activin A could be replaced by exogenous TGFβ1, we tried to conduct DE differentiation in the presence of exogenous TGFβ1. In these experiments TGFβ1 could induce a partial EMT (such as formation of F-actin and downregulation of CDH1) but the cells failed to upregulate CDH2 or SOX17 (Figure 4d). So, Activin A remains an essential component to drive DE differentiation and TGFβ appears to be one of the downstream signaling pathways that must be activated by Activin A.

5. To test whether the activation of SNAI1 is via the smad3/2/4 complex, the authors should use SIS3 inhibitor that inhibits Smad3 phosphorylation instead of RepSox. In addition, the authors should test the levels of Smad3 phosphorylation following RepSox treatment by western blot.

Response: We tried to test whether the activation of SNAI1 is via the SMAD2/3/4 complex by using SMAD3 phosphorylation inhibitor SIS3. We found that a high dosage (1 μM) of SIS3 leads to massive cell death, while SIS3 at a lower dosage of 500 nM is not able to effectively block the phosphorylation of SMAD3 and DE differentiation (see enclosed figure below, Scale bar corresponds to 20 μm.)

We tested the protein levels of SMAD2 and SMAD3 following RepSox treatment by western blot. To our surprise, we found that the total protein levels of smad2 and smad3 are significantly decreased upon RepSox treatment (see enclosed figure below), suggesting that RepSox is indirectly regulating SMAD2 and SMAD3 expression.

6. To test whether SNAI1 mediates the effect of TGF β 1, the authors should show that SNAI1 overexpression can rescue the negative effect of RepSox.

Response: To test whether SNAI1 mediates the effect of TGF β 1, we established a SNAI1-overexpressing hESC cell line. SNAI1-overexpression failed to rescue the inhibitory effect of RepSox on DE formation, however, it did rescue the Activin A induced degradation of CDH1 in the presence of RepSox (Figure 7b) as well as the formation of F-actin (see attached figure below). Thus, SNAI1 mediates parts of the TGF β 1 induced signaling during DE formation.

7. The authors should discuss whether hepatocyte differentiation from ESCs indeed requires EMT-MET process or whether ESC differentiation just mimicking embryo development and DE cells must migrate along the developing embryo to locate the progenitor cells in the correct location in the embryo.

Response: yes, we revised the manuscript to discuss the impact of EMT-MET in cell fate conversions in vitro as well as its implication in vivo (such as EMT and differentiation during gastrulation).

Reviewer #2 (Remarks to the Author):

In the present ms., the authors investigate transient EMT-MET processes in cell fate decisions, in particular during differentiation of hepatocytes from hESCs. The study is based on previous data from the authors and other groups showing that MET is required for efficient reprogramming of MEFs (Liu et al, 2010; Samavarchi-Tehrani et al., 2010). Also, and of particular interest for the present study is the previous report of the authors showing that sequential delivery of OKMS factors to MEFs (and other mouse cell lines) induce a sequential EMT-MET process driving reprogramming more efficiently than simultaneous addition of the four factors (Liu et al., 2013). Now, the authors hypothesize that similar sequential EMT-MET process might occur during cell fate decisions such as in hepatocyte differentiation. They approach this interesting question by mapping expression patterns during hepatocyte differentiation from hESCs by bulk RNA-seq. Based in this analysis they observed a synchronous EMT followed by a less synchronous MET. They then perform single cell analysis to confirm the bulk RNA-seq data and further explore the action of TGF β and the requirement of SNAI1 for EMT at initial differentiation stages. The authors provide convincing evidence for early and synchronous EMT process mediated by TGF β at early stages of hESCs differentiation (i.e., Definitive Endoderm), apparently through SNAI1. However, the study fails sort in providing robust evidence for the subsequent MET process proposed to occur for hepatoblast and hepatocyte differentiation. In addition, the study lacks robust statistical analyses required in order to support strong conclusions. On the other hand, the results showing here are a little bit striking when compared to the previous data on EMT-MET during iPS reprogramming, as in both distinct and apparently opposite biological process (reprogramming vs differentiation) the spatio-temporal sequence of events are similar: i.e., EMT followed by MET. It is also unclear if the present data can be extended to other cell fate scenarios distinct from hepatocyte differentiation. Therefore, although the present study provides data of interest to a broad scientific community, further studies are required to provide more convincing evidence to fully support that a sequential EMT-MET process occurs in cell fate decisions during differentiation.

Response: We appreciate that reviewer's comment for the EMT part of the dataset. In fact, as we hypothesized, an EMT is the critical part of our paper and should provide enough evidence to prove our EMT-MET hypothesis at this stage. The conversion of hESCs to hepatocytes-like cells is a process of changing one epithelial cell to another, and intuitively should not require an M phase, i.e., the mesenchymal fate.

We chose the ES-hepatocyte model to prove our EMT-MET hypothesis based on that consideration. There are only two possibilities: 1) hESCs simply change from being a super-epithelial cell to a hepatocyte without going through a mesenchymal phase; or 2) hESCs first become a fibroblast-like cells before assuming the epithelial hepatocytes. In the first case scenario, one would not be able to detect any EMT. If we see a EMT, it must be the second case. Following this argument, we proceeded to establish the hepatocyte differentiation protocol based on published reports (this is not our work and we do not claim any credit on this). We merely wished to see if the differentiation goes through an M phase as a result of EMT. So we were very pleased to see that the reviewer was happy with our EMT part of the story.

It is a common problem in the differentiation of hESCs to achieve mature cell types in vitro. Indeed, we believe that the less synchronous MET is part of this problem, and is one of the reasons we and others failed to derive fully mature hepatocytes. Nevertheless, we did observe that some of the cells become E-cadherin (CDH1) positive and even acquire the ability to secrete albumin-a hallmark feature of mature hepatocytes. However, as yet we cannot achieve an asynchronous MET which may impact the ability to derive fully mature hepatocytes. It was not intention to claim that our hepatocytes are mature; we wanted to observe the sequential EMT-MET process, which is what we found. We hope that the lack of fully mature hepatocytes in this study does not undermine our exploration of the sequential EMT-MET that we observe. Indeed, we believe the observation that the later MET is asynchronous may inspire others to develop new protocols for the optimized generation of hepatocytes.

Main points.

1. The authors present evidence for an EMT process occurring at initial stages of differentiation associated to definitive endoderm (from D0 to D3) based on gene expression and marker analyses. They also claim that a subsequent, although asynchronous, MET process occurs concomitant to hepatoblast and hepatocyte differentiation (from D7 to D13-D21) (Figs. 1 and 2). Detection of the MET process is mainly based in re-expression of CDH1 and some other epithelial markers (OCLN) and downregulation of CDH2 and SNAI1. However, a close inspection of gene expression in bulk RNA analyses (Fig. 1c) indicates that expression of many mesenchymal genes (VIM, FN1) and EMT-TFs (ZEB1, SNAI2, TCF4) remain at high level at D13-D21 suggesting that not a true MET process is taking place during differentiation. In my opinion, this is an important point that the authors should further investigate in order to make more strong conclusions on the relevance of the proposed MET process. Apart from

expression of hepatocyte markers the authors do not provide strong evidence for the functionality of hepatoblast and hepatocytes obtained at D13 and D21, respectively. Further characterization of the protein expression level and, importantly, localization of the above mentioned mesenchymal genes and EMT-TFs should be performed.

Response: As we mention above, the second MET is asynchronous, indeed, our single-cell qPCR data clearly supports that lack of synchronicity as by day 7 some cells begin to express CDH2 but many still express CDH1.

To expand on this point, and to confirm whether the MET process took place at later differentiation time points, we have now carried out immunofluorescence staining on day 13 and day 21 cells (Figure 2a and attached figure below). The results indicated that both AFP-positive cells were ZO-1 positive, VIM and SNAI2 negative. Meanwhile, those cells remained CDH2 positive, indicating that they have not finished their MET. The ALB-positive cells at day 21 were positive for ZO-1 and CDH1, negative for CDH2, VIM, SNAI2 and ZEB1, indicating a more complete MET in those cells. As mentioned above, exactly why these cells fail to complete the MET may ultimately be related to the lack of maturity in the in vitro-derived hepatocytes.

In order to test the functionality of 21d hepatocytes, we detected albumin and urea secretion by Elisa (supplementary figure 1b and c). The results showed that our 21d hepatocytes were able to secrete albumin and urea at levels comparable to those from the original reports (Reference #8 and 9).

2. In direct relation to the above point, single cell RNA expression analyses also broadly confirm EMT at early points (Fig. 3a). Indeed, downregulation of CDH1 and upregulation of CDH2, and other mesenchymal genes, is clearly detected in individual cells at D3. However, while upregulation of CDH1 starts to be detected at D5 it coexists with apparently high expression levels of CDH2 and other mesenchymal genes up to D7, indicating intermediate mesenchymal/epithelial states. The authors should at least discuss on this important data.

Response: We thank the reviewer for pointing out this phenomenon. Indeed, at day 7 some cells do indeed co-express both CDH1 and CDH2 at the same time, indicating an intermediated state of MET. We notice that this intermediate stage can last up to day 13, when the AFP positive cells were both positive for CDH2 (Fig. 2a) and ZO-1 (see the

above enclosed figure related to our response to point 1). Furthermore, those cells were highly motile (Fig. 2c). These observations indicate that the MET during hepatic differentiation is a slow and less efficient process when compared to the proceed EMT.

3. Strikingly, upregulation of SNAI1 at D3 at single cell level is hard to detect in the heatmap shown in Fig.3a. A similar situation occurs for other EMT-TFs, ZEB1 and SNAI2, making difficult to link the apparent EMT to expression of EMT-TFs likely because of their low expression levels. To clarify this relevant issue, relational network and scatter plots correlating the expression of CDH1 and CDH2 with SNAI1, and other EMT-TFs, should be included as those presented for CDH1, CDH2 and with DE selected markers (Fig. 3b-d).

Response: Unfortunately, we find that the SNAI1 primer did not perform well in the single-cell qPCR experiments, and we could not get good data from this gene. Looking at the violin plot for its expression (see enclosed figures below), you can see that SNAI1 has a low dynamic range (compare, for example, SOX17) and does not reach a high maximum. Single cell qPCR remains a challenging technique and primers which work well in bulk qPCR can often fail for unknown reasons when included in the single cell procedure. If the reviewer thinks it is appropriate we can remove the SNAI1 primer from the analysis, but as we use a systematic quality control scheme derived from Buganim et al., 2010, we do not think it appropriate to arbitrarily remove further genes from the analysis. Currently, although we include SNAI1 in the larger heatmap we do not highlight its expression in the relational networks, correlation heatmap or dot-plots (Figures 3b, c, d). Unfortunately, this problem was shared by all of the EMT-TFs in the single cell qPCR. Hampering their usage for this analysis.

In order to evaluate their potential roles in EMT and DE formation, we performed loss-of-function analysis on those EMT-TFs. So far, we have successfully established H1 mutant cell lines deficient in SNAI1, SNAI2, ZEB1 or ZEB2 (Supplementary figure 3). We tested the capacities of

those cells to initiate an EMT and differentiate into the DE lineage. As shown in Figure 5, SNAI2, ZEB1 or ZEB2 mutant cell line could be successfully differentiated into DE as well as hepatocytes. On the other hand, SNAI1 mutant failed to initiate part of the EMT program (such as downregulation of CDH1) and DE formation (Figure 5 and 6). These results indicate that SNAI1 is the most prominent of the EMT-TFs involved in DE formation.

4. The authors provide evidence that TGF β mediates the initial EMT (Fig. 4) apparently through induction of SNAI1. To provide evidence of Snail1 involvement in initial EMT they perform knockdown assays with siRNAs to different EMT-TFs (Fig. 5a, b), claiming that SNAI1 is the most critical EMT-TF for upregulation of CDH2 as well as of DE markers (Fig. 5b). They further show that SNAI1 knockdown blocks EMT and mesoderm/endoderm marker expression (Fig. 5c). However, the data presented in Fig. 5 lack rigorous statistical analysis that is absolutely required to support this important conclusion. In addition, the effect of SNAI1 in suppressing EMT (i.e, upregulation of CDH1, downregulation of CDH2) needs to be confirmed at the protein and phenotypic level. IF analysis should be performed to confirm the data obtained by qPCR analyses. In addition, migration assays similar to those presented in Fig. 2f, g, should be performed in the presence of SNAI1-siRNA and/or RepSox. These studies will provide more convincing evidence for the requirement of SNAI1 in the initial EMT process mediated by TGF β .

Response: We have established Knockout H1 cell lines to address the roles of EMT-TFs during DE formation. Among SNAI1/2 and ZEB1/2, SNAI1 was the only EMT-TF required for DE formation (Figure 5 and 6). SNAI1 knockout cells failed to effectively downregulate CDH1 at mRNA and protein levels and those cells were less motile at D3 (Figure 6). Meanwhile, DE markers such as FOXA2, GSC, EMOES, LGR5 (mRNA level) and SOX17 (mRNA and protein level) failed to be stimulated in SNAI1 knockout cells (Figure 5 and 6). On the other hand, SNAI2, ZEB1 or ZEB2 knockout cells were able to be induced into ALB-positive hepato-like cells at D21 (Figure 5c). These results strongly suggest that SNAI1 but not SNAI2 or ZEB1/2 as the prominent EMT-TF required for endodermal differentiation of hESCs.

5. The implication of EMT-TFs (SNAI2, TWIST1/2, ZEB1) in the initial EMT, mediated by TGF β , is discarded by the authors based in the apparently low effect of the corresponding siRNAs to repress CDH2 and DE marker induction (Fig. 5 a, b). However, as mentioned above these data lack robust statistical analyses to fully discard the effect of those siRNAs. Indeed, upregulation of SNAI2 and ZEB1 at D3 occurs at similar levels than SNAI1 and all three factors are inhibited to a similar extent by the TGF β inhibitor RepSox, as detected in

bulk qPCR analyses (Fig. 4f). These data, in principle, support the implication of at least SNAI2 and ZEB2 in early EMT and need to be further studied by more extended knockdown assays.

Response: Among the SNAI, ZEB and TWIST family EMT-TFs, SNAI1 showed the highest expression level at D3. Our knockout studies further revealed SNAI1 as the most critical EMT-TF involved in DE formation (please refer to our response to point 4 and 5). We are working on establishing TWIST1/2 knockout cell lines now and will functionally test their role in DE formation once they are available. Meanwhile, those EMT-TFs may have lineage specific functions during the targeted differentiation of hESCs. We are actively working on those possibilities.

6. In the same line, and related to comments on point 1, the potential role of SNAI2 and ZEB1 in late states of differentiation should also be tested. Is there any effect of knocking down these factors in the commitment to differentiation states (D13, D21)?

Response: We now provide evidence that SNAI2 and ZEB1 Knockout H1 can be induced into AFP positive (D13, see attached figure below) and ALB positive (D21, Figure 5c) cells as efficiently as the wild type H1 cells. These results indicated that SNAI2 and ZEB1 were not essential for late stage hepatic differentiation as well.

7. It is also striking that apparently similar spatio-temporal EMT-MET process occurs during cell fate (i.e., hepatocyte differentiation) and during reprogramming, as previously shown by the authors and others. It is also unclear if this situation is specific to hepatocyte differentiation or can be extended to other cell fate process. The authors should at least discuss on this relevant issue and present arguments on this apparent paradox.

Response: We appreciate this insightful comment from the reviewer.

This is indeed why we set out to test the EMT-MET hypothesis during differentiation in the first place and wish to propose that this is a mechanism that can unify both reprogramming and differentiation, although the details for the regulators may be quite different. We revealed an EMT-MET during hepatic differentiation of hESCs and SNAI1 as a major regulator of the initial EMT. On the other hand, the initiation of neuroectoderm differentiation did not undergo a classic EMT and SNAI1 was not required for the process (Figure 8). Further studies are required to determine the lineage or stage specific roles of EMT during targeted differentiation of hESCs other lineages.

8. Statistical analyses need to be included in all relevant Figures (Fig. 2a-c, g; Fig. 4f, h; Fig. 5a-c)

Response: We appreciate this comment from the reviewer. We performed statistical analysis and revised the text, figures and figure legends accordingly.

Minor points.

1. Mention to Suppl. Fig S1 is lacking in the text.

Response: We have amended the text to correct this error.

2. The different gene categories indicated in the legend to Fig. 1c should be labeled in the Figure panel to ease its reading to non-specialists in hepatocyte or EMT fields.

Response: As suggested, we have added labels to help guide the reader.

Reviewer #3 (Remarks to the Author):

In the present study Duanqing Pei and colleagues report evidence for a sequential EMT-MET process during hESC differentiation towards hepatocytes. As an initial experiment they performed in vitro differentiation of human ESCs into hepatocytes and performed bulk RNA-seq experiments throughout the whole differentiation time course. To further analyze heterogeneity of the differentiating cell population they conducted single cell qPCR analysis on 46 relevant genes for early time points of the differentiation process. Some of the findings were further investigated by immunohistochemistry and siRNA mediated knockdown experiments. The observation of a sequential EMT-MET process during differentiation of pluripotent stem cells into mature epithelial cell

types as a general phenomenon would be very interesting and relevant to a broader audience. However, in the current study only a specific system is analyzed and the generality of the finding is unclear. To prove the generality at least one more differentiation system should be analyzed.

Response: We would like to thank the reviewer for constructive assessment of our study. As the reviewer indicates, we were perhaps a little overambitious in attempting to generalize the EMT-MET paradigm to multiple differentiation systems. Consequently we have elected to restrict the study to the endoderm, and have rewritten the title and manuscript to address a more limited scope. We continue to discuss how this EMT-MET model might apply to other systems, but limit ourselves to speculation in the discussion.

Moreover, although the study provides in my opinion clear evidence for an initial epithelial-to-mesenchymal transition, the subsequent mesenchymal-to-epithelial transition is not apparent from the data (see major concerns below). Therefore, the major conclusion of the study is not sufficiently supported by the data presented in the manuscript, and additional experiments and data analysis are required.

Response: we really appreciate this comment which was also raised by reviewer 2. Again, we would be willing to limit our scope of conclusion and emphasize just the fact that an M phase occurs during DE differentiation. We appreciate that reviewer's comment for the EMT part of the dataset. However, we respectfully disagree that the cells do not undergo a second MET. Hepatocytes are an epithelial cell type, that express CDH1 and have low levels of CDH2 (see our new Fig 1c for RNA-seq data from primary hepatocytes). Consequently, as the cells go through an observed EMT, and an EMT that is required for correct DE differentiation, it is intuitive that the cells later go through a corresponding MET to complete their differentiation. The asynchronous MET may be related to the fact that we and also others fail to derive mature hepatocytes. Nevertheless, we did observe some of the cells become CDH1 positive and even acquire the ability to secrete albumin, a hallmark feature of mature hepatocytes. However, inherent in the differentiation system, we cannot achieve a synchronous MET, which we believe is responsible for the lack of fully mature hepatocytes. This study is an exploration of the sequential EMT-MET that the cells undergo as they differentiate in vitro.

Major concerns:

1. Based on the PCA in Figure 1b it is claimed that starting from an epithelial

state cells traverse through a mesenchymal state and then revert to an epithelial state. The authors make the statement that PC3 separates an epithelial from a mesenchymal phenotype. However, neither do the authors explain how this was inferred, nor do they show any supporting data. I assume this observation is based on the loadings of epithelial and mesenchymal markers, but this should be reported in the text.

Response: We have added the gene loadings for PC1-3 in the new Supplementary figure 2. (Note that, as mentioned above, PCA is now performed using a new implementation and PC1 is now the first informative PC. No conclusions were altered due to the change in implementation). This does indeed indicate that PC2 broadly reflects an epithelial-mesenchymal state. For example see the split in the CDh1/CDH2 loading at opposite ends of the PC loading. Similarly, PC1 indicates pluripotency, PC2 acquisition of DE, and PC1 and PC3 hepatoblast and hepatocyte acquisition. However, it should be noted that the interpretation of PCA is somewhat arbitrary, and PC3 also contains indications of EMT/MET genes. PCA rarely separates biological processes cleanly, but we find that it remains a useful indication about the trajectories of the cells as they differentiate, and gives a simple representation of an otherwise very complex dataset.

2. Figure 1c and 1d show the down-regulation of epithelial genes and the up-regulation of mesenchymal genes between day 0 and day 3, it appears that mesenchymal genes such as Vimentin remain up-regulated at later timepoints. How can this be reconciled with a subsequent MET during these later timepoints?

Response: We appreciate this critical comment. As data for Figure 1c were from bulk RNAseq, we assumed that the level of those mesenchymal genes remain high due to the presence of some mesenchymal type cells in the mixture. To confirm this, we have performed immunofluorescent assays and showed that both AFP-positive and ALB-positive cells were negative to those mesenchymal markers mentioned above, except CDH2 at D13 (Figure 2a and the attached figure below). So, at least the AFP and ALB positive cells underwent MET while a fraction of cells remained mesenchymal, i.e., positive for those markers the reviewer pointed out in Figure 1c and 1d.

3. Figure 2c clearly shows up-regulation of mesenchymal genes at day 3 after normalizing to GAPDH. How highly are these genes expressed at later timepoints (day 13) after normalization to GAPDH?

Response: We appreciate this critical comment from the reviewer. We detected mesenchymal gene expression at different time points via qRT-PCR (see Figure 1d) measured relative to GAPDH. It should be noted that GAPDH is one of the top 10 highest expressed genes in cells (as measured by RNA-seq), hence the relative levels compared to GAPDH remain relatively low. The qPCR results indicated that EMT-TFs (e.g.SNAI1) and mesenchymal genes (e.g. VIM) were up-regulated at early time points and then down-regulated at later time points. This agrees with the hypothesis that an EMT-MET process happens during hepatocyte differentiation. We also noticed that SNAI2 and ZEB1 were up-regulated at later time points. This might be due to the existence of mesenchymal type cells that failed to be induced into hepato-like cells in the mixture. In addition, we now report that knockout of SNAI2, ZEB1 or ZEB2 did not disrupt the hepatic differentiation of hESCs (Figure 5). Thus, SNAI1 is *the* major EMT-TF required for the EMT and hepatic differentiation of hESCs.

4. A possible explanation for the remaining high levels of mesenchymal markers could be post-transcriptional down-regulation. In Figure 2d it is shown that no CDH2 protein is detected although it is still up-regulated on the mRNA level (Figure 2s). The authors should measure protein expression at day 13 for additional mesenchymal markers to test this hypothesis.

Response: We measured the protein levels of mesenchymal markers at D13 (Figure 2a and the enclosed figure below). The results indicated that AFP-positive cells were negative for CDH1, VIM and SNAI2, while positive for CDH2 and ZO-1, indicating an intermediate status of MET.

5. In Figure 2f and g, a migration assay is presented, proving the mesenchymal state of DE cells. These experiments should also be performed at later differentiation stages to prove a loss of the mesenchymal character.

Response: We appreciate this critical comment from the reviewer. We carried out migration assay for cells at D0, D3, D13 and D21 (Figure 2c). The results revealed that cells of D3 and D13 had high migration ability, while the more mature hepato-like cells of D21 gradually lost the migration activity. This is consistent with the results of our gene expression analysis (Figure 1d and 2a) and does indeed support a sequential EMT-MET occurring in these cells.

6. The single cell gene expression analysis nicely demonstrates that the cell-population is fairly synchronized during the first seven days of the time course. This also shows that the remaining expression of mesenchymal markers is not due to population heterogeneity. The data shown in Figure 3b and c confirm the correlated expression of CDH1 and CDH2 at day 5 and 7. The correlation between CDH1 and other mesenchymal markers should also be analyzed. More importantly, the authors should add data for day 13 or 21 to measure the expression of mesenchymal markers at a later stage.

Response: We indeed tried to draw relational network and scatter plots

correlating the expression of CDH1 and CDH2 with SNAI1 and other EMT-TFs. Unfortunately, as described above for reviewer 2, we were unable to draw a clear conclusion from those analysis, most likely due to the very low expression levels of those EMT-TFs. The expression of EMT-TFs such as SNAI2 and ZEB1 was very low at D3 (Figure 1d). SNAI1 was expressed at higher mRNA level (Figure 1d) and protein level (attached figure below) at D3, however, it was also expressed at D0, making it difficult to perform the above mentioned analysis.

In order to evaluate their potential roles in EMT and DE formation, we performed loss-of-function analysis on those EMT-TFs. We successfully established H1 mutant cell lines deficient in SNAI1, SNAI2, ZEB1 or ZEB2 (Supplementary figure 3). We tested the capacities of those cells to initiate EMT and differentiate into DE lineage. As shown in Figure 5, SNAI2, ZEB1 or ZEB2 mutant cell line could be successfully differentiated into DE as well as hepatocytes. On the other hand, SNAI1 mutant failed to initiate part of the EMT program (such as downregulation of CDH1) and DE formation (Figure 5 and 6). These results indicate that SNAI1 is the most prominent EMT-TFs involved in DE formation.

We tried but failed to perform similar single cell qPCR analysis for D13 and D21 cells (most likely due to technical problems in dealing with the extensively heterogeneous cell population). In order to measure the expression of mesenchymal markers at a later stage, we have done immunofluorescent assays for D13 and D21 cells (Figure 2a and the attached figure below). The results indicated that both AFP-positive cells were ZO-1 positive, VIM and SNAI2 negative. Meanwhile, those cells remained CDH2 positive, indicating that they have not finished MET. The ALB-positive cells at d21 were positive for ZO-1 and CDH1, negative for CDH2, VIM, SNAI2 and ZEB1, indicating a more complete MET in those cells.

7. Based on the presence of CDH2-positive cells that do not express SOX17 and the absence of SOX17 positive cells that are negative for CDH2 the authors claim that EMT precedes specification of the DE. These data are not sufficient to infer a temporal order of the events. Perhaps the SOX17-negative cells are able to undergo EMT but unable to differentiate (corresponding to the POU5F1 positive cells at day 7 shown in Figure 3b). This could be tested by co-staining pluripotency markers (see also 8.).

Response: We appreciate this critical comment from the reviewer. We have co-stained pluripotency markers with endoderm lineage marker (see attached figure below). The results showed that SOX17-negative cells were indeed positive for pluripotency markers (POU5F1, NANOG or SOX2), indicating a failure of differentiation.

8. The authors perform knockdown experiments of transcription factors involved in the EMT. Based on the data presented in Figure 5c it is apparent that knockdown of SNAI1 has a strong effect on DE specification, but only a moderate effect on the expression of CDH1 and CDH2. How does the expression of other epithelial and mesenchymal markers change? The authors should perform additional experiments (protein staining, migration assay) to demonstrate that the knockdown inhibits EMT and that this is the primary cause of failure of DE specification. However, even with these data additional experiments are required to support the claim that the EMT precedes DE specification. At the very least it should be tested if EMT still occurs in the absence of DE specification genes by knockdown experiments.

Response: We have now established Knockout H1 cell lines to address the roles of EMT-TFs during DE formation. Among SNAI1/2 and ZEB1/2, SNAI1 was the only EMT-TF required for DE formation (Figure 5 and 6). SNAI1 knockout cells failed to effectively downregulate CDH1 at mRNA and protein levels and those cells were less motile at D3 (Figure 6). Meanwhile, DE markers such as FOXA2, GSC, EMOES, LGR5 (mRNA level) and SOX17 (mRNA and protein level) failed to be stimulated in SNAI1 knockout cells (Figure 5 and 6).

We performed additional experiments to dissect the relationship between the EMT and DE specification. Firstly, we found the cytoskeletal change has been very obvious during DE differentiation (See the attached figure below). Our results showed that acute cytoskeletal changes happened before cells become SOX17-positive.

Secondly, we performed siRNA mediated knockdown of SOX17 and found that depletion of SOX17 effectively blocks the induction of DE markers such as FOXA2 but it does not affect EMT related changes (F-actin formation, induction of CDH2, SNAI1) (see the attached figure below). Together, these observations indicate that EMT occurs prior to SOX17 expression and SNAI1 is required for the induction of SOX17 but not vice versa.

Minor concerns:

- Figure 2b is not cited in the main text and Figure 2d is discussed prior to Figure 2c.

Response: We have corrected this error in the new manuscript.

2. In Figure 3b the authors show relational network plots without discussing how these were derived. What does proximity of two datapoints in this representation reflect and what does a link between two datapoints represent?

Response: The generation of the relational networks was described in the methods, briefly the data is subjected to PCA and the most informative PCs are projected into n-dimensional space. (In this case, PC's 1 though 3). The Euclidean distance (straight line) is measured, and for a particular threshold a network is drawn. The network is then positioned based upon the number of edges between each cell and closely related cells, and hence the position is representative of the relative similarity of a particular cell to other cells. Note that this method differs from methods like MDS or tSNE in that it does not have any units on the x and y-axes.

3. In Figure 3d the gene names should also be shown along the horizontal axis to ease reading of the heatmap.

Response: We appreciate this comment from the reviewer and revised the figure as suggested.

Reviewer #4 (Remarks to the Author):

Reprogramming pluripotent cells into hepatocytes is a topic of interest in the field of cell transplantation therapy and developmental biology. The manuscript investigated the EMT-MET processes of hES cells using single cell analysis. The study also identified SNAI1 as a downstream mediator of Activin A-induced differentiation. However, the manuscript is currently unacceptable due to the following reasons:

(1) Statistics: authors did not provide any information about (a) statistical significance, for their entire analyses; (b) number of replicates (biological and/or analytical), for RNA-seq analyses and single cell analyses. Therefore, it was impossible to determine whether the paper is technically sound, and whether the results are reliable.

Response: All results for qPCR analysis and migration assays in the manuscript were from at least three independent biological replicates. We performed statistical analysis when applicable and the results were present in the revised figures and text. The RNA-seq was done in singlicate, whilst by necessity the single cell qPCR experiments cannot have biological replicates and we find technical replicates offer no

utility as they correlate nearly perfectly. The major conclusions from RNA-seq and single cell experiments were further confirmed by independent qPCR and immunofluorescence staining experiments. In addition, we used knockout cell lines to demonstrate functionally that SNAI1 signaling and EMT is required for DE formation.

(2) Novelty: the fact that hES cells treated with Activin A undergo an EMT while differentiating into definitive endoderm has been already reported in the reference no. 11 :

D'Amour KA et al. Efficient differentiation of human embryonic stem cells to definitive endoderm. Nat Biotechnol. 2005 Dec;23(12):1534-41.

Authors are claiming that the present study is investigating the acquisition of a genuine mesenchymal phenotype (page 5). To support their claims, additional criteria for EMT-MET such as redistribution of ZO1 and other cytoskeletal proteins at multiple time points including day 21 need to be investigated.

Response: We were inspired by the abovementioned paper about the occurrence of EMT during the in vitro differentiation of hESCs into DE. Since cells in vitro do not seem to need to migrate to the right position to receive the differentiation signal, we wanted to know whether the EMT in vitro is just mimicking gastrulation in vivo or it is functionally required for differentiation into DE. We performed qPCR analysis (Figure 1d), immunostaining assay (Figure 2a) on EMT related genes and migration assays (Figure 2b, c) for cells at D0, D3, D13 and D21. Collectively, the results indicated that DE cells of D3 were mesenchymal like, cells at D13 were in a transitional status while the ALB-positive cells at D21 were epithelial-like. We then started to identify signals that induce EMT during DE formation and found that an Activin A stimulated TGF β secretion induces the EMT via a SNAI1 dependent pathway. We feel this study provides a novel insight into the relationship between EMT, MET and hepatic differentiation of hESCs and encourage people to investigate the regulation and function of EMT in other cell fate conversions.

(3) Conclusions: Data presented are not sufficient to support their conclusion that cells transition into a mesenchymal state at day 3 and then revert back to an epithelial state (page 4):

(a) Figure 2: to prove that cells at day 3 indeed acquired mesenchymal characteristics, authors should include appropriate controls such as mesenchymal cells (such as MEFs) and epithelial cells (such primary hepatocytes) in there analyses.

Response: We have added MEFs as a positive control for our immunostaining. Stress fibers are clearly detected in these cells (see

Figure below). **SNAI2 and ZEB1 were detected in the nuclei of these cells while CDH2 is hardly detectable in these cells. Primary hepatocytes, which would be an ideal epithelial control for this study, were very difficult for us to obtain.**

Nevertheless, a comparison of cells at D3 to D0 (which are typical epithelial cells) or D21 clearly indicated that D3 cells are most mesenchymal-like (Figure 2a) and supported an epithelial-mesenchymal-epithelial change during hepatic differentiation of hESCs.

(b) Figures 2F, 2G: it is unclear whether cells lose their migratory phenotype at later time points.

Response: We have added migration assay data for D0, D3, D13, D21 cells (Figure 2b, c). The results revealed that cells of D3 and D13 had high migration ability, while the more mature hepato-like cells of D21 gradually lost the migration activity. This is consistent with the results of our gene expression analysis (Figure 1d and 2a)

(c) Expression of some of mesenchymal markers is sustained throughout the differentiation process until day 21 (Figures 1C, 1D)

Response: We appreciate this comment which has also been raised by previous reviewers. As detailed above, we adopted the protocols from the literature and realized that the differentiation process was not synchronous for all cells, especially during the later MET phase. As such, there were cells resistant to the later MET process, and thus remain in a mesenchymal state. We think it is interesting and possibly related to the lack of maturity in the cells that they are not completing the MET.

According to the bulk RNAseq data, D13 hepatoblasts and D21 hepatocytes express many mesenchymal genes (ZEB1, VIM, SNAI2, etc). In order to confirm whether the MET process took place at later differentiation, we have now carried out immunofluorescence staining on D13 and D21 cells (Figure 2a and attached figure below). The results indicated that both AFP-positive cells were ZO-1 positive, VIM and

SNAI2 negative. Meanwhile, those cells remained CDH2 positive, indicating that they have not finished MET. The ALB-positive cells at D21 were positive for ZO-1 and CDH1, negative for CDH2, VIM, SNAI2 and ZEB1, indicating a more complete MET in those cells.

Reviewers' Comments:

Reviewer #1 (Remarks to the Author)

The reviewer is satisfied with the revised version

Reviewer #2 (Remarks to the Author)

The authors have satisfactorily addressed the main concerns by performing additional experiments and extensive modification of the ms. The required statistical analyses have also been included. I congratulate the authors for the effort and for providing robust evidence to their main hypothesis

Reviewer #3 (Remarks to the Author)

The authors have addressed all my concerns in the revised manuscript. They convincingly demonstrated EMT-MET for endodermal differentiation from hESCs induced by Activin A. They identified SNAI1 as a key factor for differentiation of this lineage and could now show that EMT is a required step preceding DE specification.

Reviewer #4 (Remarks to the Author)

The study elucidates novel molecular mechanisms underlying EMT and the formation of DE from embryonic stem cells using single cell analyses and loss-of-function studies, and is therefore highly significant. While the revised manuscript has been substantially improved, additional data are required to support the statement that "a sequential EMT-MET process drives the hepatic differentiation of hESCs" (discussion page 20). The vast majority of data (figures 4-7) still focus at the early EMT stages. It still remains unclear how treatment with Repsox at early stages and genetic ablation of SNAI1 affect hepatocyte markers at day 21. The Authors should at least demonstrate that Repsox treatment and ablation of SNAI1 block the expression of albumin at day 21 using immunofluorescence staining. Furthermore, while data support the conclusion that SNAI1 is required for Activin A-induced DE formation and modulation of CDH1 expression, further discussions are required on what regulates expressions of CDH2, vimentin, and F-actin formation (Figures 5 and 6). The Authors should also discuss potential regulators of subsequent MET during the later stages.

(Minor) For Figure 2, it is recommended to include D0 cells as a negative control for Sox17, AFP, and staining.

Reviewer #4 (Remarks to the Author):

The study elucidates novel molecular mechanisms underlying EMT and the formation of DE from embryonic stem cells using single cell analyses and loss-of-function studies, and is therefore highly significant. While the revised manuscript has been substantially improved, additional data are required to support the statement that “a sequential EMT-MET process drives the hepatic differentiation of hESCs” (discussion page 20). The vast majority of data (figures 4-7) still focus at the early EMT stages. It still remains unclear how treatment with Repsox at early stages and genetic ablation of SNAI1 affect hepatocyte markers at day 21. The Authors should at least demonstrate that Repsox treatment and ablation of SNAI1 block the expression of albumin at day 21 using immunofluorescence staining.

Response: First of all, we appreciate the effort from the reviewer to make our manuscript much better and also would like to thank the reviewer for a positive recommendation. In regard to the specific point raised here, we could have done that easily given the amount of efforts we have devoted to this entire project. However, in our chemically defined differentiation system, the SNAI1^{-/-} cells appear OK at D3, but they gradually died during the second stage of induction (FGF4+BMP2). Similarly, we also observed cell death in the Repsox treated samples. So with much regret, we have not been able to assay the induction of ALB in those cells. We will keep this question in mind in our ongoing work and see if we can test this as suggested by the reviewer and hopefully we can report the results in our future publications.

Furthermore, while data support the conclusion that SNAI1 is required for Activin A-induced DE formation and modulation of CDH1 expression, further discussions are required on what regulates expressions of CDH2, vimentin, and F-actin formation (Figures 5 and 6).

Response: We appreciate this comment and have addressed this in the discussions. Since Repsox is able to block the Activin-A induced upregulation of

CDH2 and vimentin as well as F-actin formation, TGF β signaling is required for those changes. SNAI1 is best characterized by its activity to suppress CDH1 expression during TGF β -induced EMT. In our DE differentiation protocol, SNAI1 knockout appeared not to affect the induction of CDH2 (Fig. 5C). The same is true in the neural differentiation (see the attached figure below). We also know that SNAI2, ZEB1 or ZEB2 is not essential for the induction of CDH2 (Fig. 5C). It is possible that other EMT-related transcriptional factors (for example, TWIST1/2, KLF8, FOXC2 etc.) are involved in the induction of CDH2 and F-actin. It is also possible that the above-mentioned factors play redundant roles in the process so deficiency in one of them is not sufficient to result in CDH2 or F-actin related defect.

The Authors should also discuss potential regulators of subsequent MET during the later stages.

Response: We appreciate this comment and provided discussion on this.

HNF4A is transcriptional factor required for the formation of hepatic epithelium and liver architecture in mouse (Parviz 2003, PMID: 12808453) and overexpression of HNF4A is able to promote MET and hepatic maturation of hESC derived hepatoblasts (Takayama 2012, PMID: 22068426). Furthermore, it is one of the critical factors used in the direct conversion of mouse or human

fibroblasts to hepatocytes (Sekiya 2011, PMID: 21716291; Huang 2014, PMID: 24582927). Interestingly, we noticed that HNF4A is highly stimulated at D13 after the treatment of HGF+KGF (Fig. 1a). We propose that the upregulation of MET factor such as HNF4a together with the downregulation of EMT factor such as SNAI1 (Fig. 1d) are responsible for the MET phase of our hepatic differentiation.

(Minor) For Figure 2, it is recommended to include D0 cells as a negative control for Sox17, AFP, and ALB staining.

Response: Yes, we will include the suggested staining as negative control in the revised version of Figure 2.

Reviewers' Comments:

Reviewer #4 (Remarks to the Author)

The Authors have addressed all the comments.